# Chroma-VAE: Mitigating Shortcut Learning with Generative Classifiers

**Wanqian Yang**    **Polina Kirichenko**    **Micah Goldblum**    **Andrew Gordon Wilson**

New York University
{wanqian, pk1822, goldblum}@nyu.edu, andrewgw@cims.nyu.edu

## Abstract

Deep neural networks are susceptible to shortcut learning, using simple features to achieve low training loss without discovering essential semantic structure. Contrary to prior belief, we show that generative models alone are not sufficient to prevent shortcut learning, despite an incentive to recover a more comprehensive representation of the data than discriminative approaches. However, we observe that shortcuts are preferentially encoded with minimal information, a fact that generative models can exploit to mitigate shortcut learning. In particular, we propose Chroma-VAE, a two-pronged approach where a VAE classifier is initially trained to isolate the shortcut in a small latent subspace, allowing a secondary classifier to be trained on the complementary, shortcut-free latent subspace. In addition to demonstrating the efficacy of Chroma-VAE on benchmark and real-world shortcut learning tasks, our work highlights the potential for manipulating the latent space of generative classifiers to isolate or interpret specific correlations.

## 1 Introduction

As science fiction writer Robert Heinlein quipped in his 1973 novel *Time Enough for Love*, progress is made not by "early risers", but instead by "lazy men trying to find easier ways to do something". Indeed, we can accuse modern machine learning models of emulating the same behaviour. There are notable examples of models that wind up learning the wrong things. For example, images of cows standing on anything other than grass fields are commonly misclassified because the combination of cows and grass fields is so prevalent in training data that the model simply learns to rely on the background as a predictive signal [4]. A more concerning example involves predicting pneumonia from chest X-ray scans. Hospitals from which training data is collected have differing rates of diagnosis, a fact that the model easily exploits by learning to detect hospital-specific metal tokens in the scans rather than signals relevant to pneumonia itself [52].

Deep neural networks can learn brittle, unintended signals under empirical risk minimization (ERM) [11, 16, 13], a well-known phenomenon observed and studied by various communities. These signals often possess two key attributes: (i) they are *spuriously correlated* with the label and are therefore strongly predictive [6, 50, 31], despite having no meaningful semantic relationship with the label, and (ii) they are learnt by the neural network as a result of its *inductive biases* [43, 10]. A recent unifying effort by Geirhos et al. [10] coins the term *shortcut* to describe such a signal. Networks that learn shortcuts fail to generalize to relevant or challenging distribution shifts.

Prior work has sought to alleviate this problem under various formalizations, most commonly in the settings of group robustness [e.g. 39, 28, 7, 25, 20] or adversarial robustness [e.g. 5, 45]. In this paper, we motivate a different approach to mitigate undesirable shortcuts, where we seek instead to learn a *shortcut-invariant representation* of the data — that is, "everything but the shortcut".

36th Conference on Neural Information Processing Systems (NeurIPS 2022).

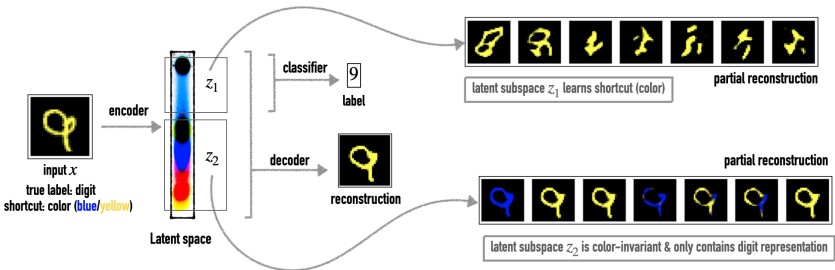

Figure 1: A visual representation of Chroma-VAE. The latent space is partitioned into two subspaces, one of which is incentivized to encode the shortcut under ERM. The complementary subspace learns a shortcut-invariant representation.

To this end, we first observe empirically that deep neural networks preferentially encode shortcuts as they are the most efficient compression of the data that is strongly predictive of training labels. This observation helps to explain why shortcut learning is so prevalent amongst discriminative classifiers.

However, this same preference can be exploited in a *generative* classifier to learn a shortcut-invariant compression. Specifically, we consider the Variational Auto-Encoder (VAE) [23], capable of learning latent representations of the data. Our key insight is to back-propagate the classifier's loss through a latent *subspace*, while reconstructing with the *entire latent space*. The model minimizes classification error by encoding the shortcut in this subspace. Since the shortcut representation is isolated, the complementary subspace is free to learn a shortcut-invariant representation through reconstruction. The final classifier is then trained on this invariant representation. We dub our approach **Chroma-VAE**, inspired by the technique of chromatography, which separates the components of a chemical mixture travelling through the mobile phase. Figure 1 summarizes our approach. We note that our pipeline is not in principle restricted to VAEs, and could be used in conjunction with other families of generative models to compartmentalize a latent representation into a shortcut and semantic structure.

Our contributions are as follows:

- We show empirically that: (i) deep neural networks preferentially encode shortcuts as the most efficient compression (Section 3.1), and (ii) the shortcut representation can be sequestered in a latent subspace of a VAE classifier (Section 3.2).
- These observations allow us to propose Chroma-VAE (Section 3.3), an approach designed to learn a classifier on a shortcut-invariant representation of the data. We demonstrate the effectiveness of Chroma-VAE on several benchmark datasets (Section 5).

Our code is publicly available at .

## 2  Background and Notation

Even though shortcuts are present in many domains within deep learning [4, 50, 2, 34, 30, 12], we restrict our discussion to image classification tasks, as (i) VAEs are widely applied to modeling image distributions, and (ii) shortcut learning is ubiquitous in the vision domain.

Let $\mathcal{D}_{tr} = \{\mathbf{x}_i, y_i\}_{i=1}^{N}$ denote i.i.d. training data. A variational auto-encoder (VAE) [23] $(E_\phi, D_\theta)$ models $\mathbf{x}$ with a latent variable: $p(\mathbf{x}, \mathbf{z}) = p(\mathbf{z})p(\mathbf{x}|\mathbf{z})$. The prior $p(\mathbf{z})$ is typically $\mathcal{N}(\mathbf{0}, I)$ and the likelihood $p(\mathbf{x}|\mathbf{z})$ is implicitly modeled by the decoder network $D_\theta(\mathbf{z}) = \left(\mu_\theta(\mathbf{z}), \mathrm{diag}\left(\sigma_\theta^2(\mathbf{z})\right)\right)$ as $\mathcal{N}\left(\mathbf{x}|\mu_\theta(\mathbf{z}), \mathrm{diag}\left(\sigma_\theta^2(\mathbf{z})\right)\right)$. Maximizing $p(\mathbf{x})$ directly is intractable due to the integral over $\mathbf{z}$. Instead, we use an encoder network $E_\phi(\mathbf{x}) = \left(\mu_\phi(\mathbf{x}), \mathrm{diag}\left(\sigma_\phi^2(\mathbf{x})\right)\right)$ as an (amortized) variational approximation $q_\phi\left(\mathbf{z}|\mu_\phi(\mathbf{x}), \mathrm{diag}(\sigma_\phi^2(\mathbf{x}))\right)$ and maximize the Evidence Lower BOund (ELBO):

$$\mathrm{ELBO}(\theta, \phi, \mathbf{x}) := \mathbb{E}_{\mathbf{z} \sim q_\phi}\left[\log \frac{p_\theta(\mathbf{x}, \mathbf{z})}{q_\phi(\mathbf{z}|\mathbf{x})}\right] \tag{1}$$

by computing unbiased estimates of its gradients.

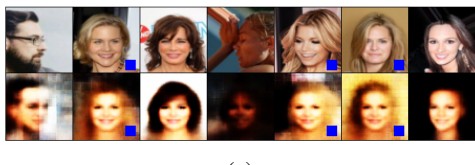 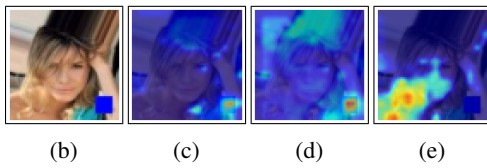

(a)                 (b)  (c)  (d)  (e)

Figure 2: **(Left) (a)** Examples of reconstructions by a decoder trained independently on the discriminative model's hidden layer. The shortcut (blue square) is clearly visible when it is present. Reconstruction is particularly poor on images *with* the shortcut — the reconstructed faces seem to collapse to the same archetype that does not resemble the original images. **(Right)** Grad-CAM heatmaps on a `CelebA` test image. **(b)** The original image with the shortcut patch. **(c)-(e)** Averaged activations across all training epochs on three separate models, with a bottlenecked hidden layer of sizes 4, 32, and 128 respectively. The smaller the bottleneck, the more likely it is that the model only focuses on the shortcut patch. With more capacity, the model focuses on other regions of the image.

In a hybrid VAE classifier $(E_\phi, D_\theta, C_\varphi)$, we model $\mathcal{D}_{tr}$ as $p(\mathbf{z}, \mathbf{x}, y) = p(\mathbf{z})p(\mathbf{x}|\mathbf{z})p(y|\mathbf{x})$. $p(y|\mathbf{x})$ is modeled by a classifier network $C_\varphi$ that takes the latent mean $\mu_\phi(\mathbf{x})$ as input and outputs class probabilities, i.e. $p_{\varphi,\phi}(y|\mathbf{x}) = \text{CE}\Big(y, C_\varphi\big(\mu_\phi(\mathbf{x})\big)\Big)$, where CE is the cross-entropy loss. Again, maximizing $p(\mathbf{x}, y)$ directly is intractable and the training objective becomes:

$$\mathcal{L}(\theta, \phi, \varphi) := -\sum_{(\mathbf{x},y)\in\mathcal{D}_{tr}} \text{ELBO}(\theta, \phi, \mathbf{x}) + \lambda \cdot \log p_{\varphi,\phi}(y|\mathbf{x}) \qquad (2)$$

where $\lambda$ is a scalar multiplier widely used in practice to account for the fact that $p(\mathbf{x})$ is typically magnitudes smaller than $p(y|\mathbf{x})$ on high-dimensional image datasets.

**Group Terminology** Let $s \in \mathcal{S}$ denote the shortcut (spurious) label. The *group* of an input $\mathbf{x}$ is $g = (s, y)$, the combination of its shortcut and true labels. *Majority groups* refer to examples with the dominant correlation between $s$ and $y$ on the training data, while *minority groups* refer to the small number of examples with the opposite correlation that ERM models typically misclassify. *Group robustness* refers to the ability of models to generalize to distribution shifts where shortcuts are no longer predictive. Methods to improve group robustness can make use of group annotations; however, our method *does not assume access* to group labels at train or test time.

## 3 Chroma-VAE: Separating Shortcuts Generatively

The key intuition behind our approach is to exploit shortcut learning for representation learning, by using a VAE to learn a shortcut-invariant representation of the data. Two central ideas, supported by empirical observations, motivate our method: (i) deep neural networks preferentially encode shortcuts under finite and limited representation capacity, and (ii) the shortcut representation can be sequestered in a latent *subspace* of the VAE when jointly trained with a classifier.

### 3.1 Shortcuts are preferentially encoded under limited representation capacity

Why do shortcuts exist? As Geirhos et al. [10] note, shortcuts arise because the model's inductive biases (the sum total of interactions between training objective, optimization algorithm, architecture, and dataset) favour learning certain patterns over others; e.g., convolutional neural networks (CNNs) prefer texture over global shape [3, 9].

We consider shortcuts from an information-theoretic perspective. Deep neural networks are commonly thought of as representation learners that optimize the information bottleneck (IB) trade-off, i.e. they aim to learn a maximally compressed representation (minimally sufficient statistic) that fits the labels well [47, 42]. Our key empirical observation is that in datasets where shortcuts exist, **they are often efficiently compressed, and the compressed information is predictive of labels**. As such, deep models preferentially encode shortcuts, especially under limited representation capacity.

We consider an experiment on the `CelebA` dataset [29], where the task is predicting hair color ("blonde" or not). We inject a synthetic shortcut in the form of a $10 \times 10$ blue "patch", superimposed

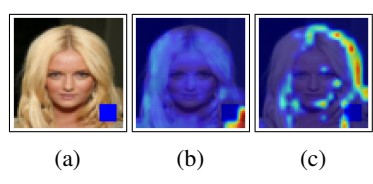
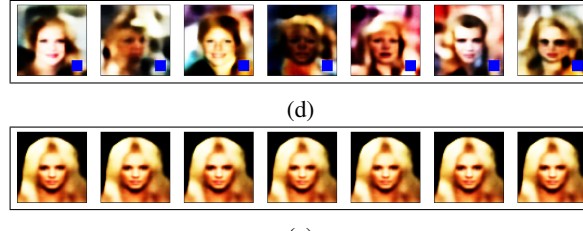

(a)   (b)   (c)

(d)

(e)

Figure 3: **(a)** Original image containing shortcut. **(b)** Grad-CAM with $\mu_\phi(\mathbf{x})_1$ gradients. **(c)** Grad-CAM with $\mu_\phi(\mathbf{x})_2$ gradients. **(d)** Partial reconstructions $\tilde{\mathbf{x}}_1$. **(e)** Partial reconstructions $\tilde{\mathbf{x}}_2$. While no patches are present in samples of $\tilde{\mathbf{x}}_2$ here, we have observed that they can occur with small probability. Conversely, across our many repeated samplings, the shortcut patch is *always* present in samples of $\tilde{\mathbf{x}}_1$. Note also that $\tilde{\mathbf{x}}_1$ samples contain a variety of hair colors, whereas $\tilde{\mathbf{x}}_2$ samples faithfully reconstruct the celebrity's blonde hair.

onto the image with 0.9 probability for the positive class (see Figure 2b). We first train a discriminative classifier on the labels, and then *independently* learn a decoder on the (fixed) hidden layer of the trained discriminative model.

Figure 2a shows the reconstructions of the decoder. The shortcut is accurately replicated in sharp detail where it exists, but other regions of the reconstructions are not as faithful to the originals. In particular, note that reconstructions are even less accurate for examples of the positive class (with the shortcut), where mode collapse has happened. Since the decoder was learned separately, this observation implies that the classifier's latent compression primarily encoded for the shortcut's presence.

This result is further supported by visualizing the activated regions of the input images during training. We implement Grad-CAM [41], which computes a linear combination of activation maps weighted by the gradients of output logits (or any upstream parameters), to produce a heatmap superimposed on the original image, showing regions of the image with the greatest positive contributions to the network's prediction.

In Figures 2b-e, we visualized these heatmaps on models with a bottleneck hidden layer of different sizes. As we can see, the smaller the bottleneck size, the stronger the activations on the shortcut patch. With additional model capacity, the model is more likely to encode other relevant (or spurious) features *in addition to* the shortcut itself.

From these experiments, we see that deep neural networks tend to compress input data in a shortcut-preserving manner when such shortcuts are both (i) compressible with little information and (ii) predictive of training labels. This tendency is exacerbated under limited latent capacity and is the reason why discriminative classifiers are susceptible to shortcut learning under ERM.

### 3.2 VAE classifiers can sequester shortcuts into specific latent dimensions

What presents a problem for discriminative modeling can be exploited into an advantage using a generative model. The key insight here is that we can train a VAE classifier where the latent space is partitioned into two disjoint subspaces — where only *one* subspace feeds into a classifier and is backpropagated through using labels. This subspace encodes the shortcut precisely because this information maximizes the classifier's performance (i.e. simultaneously exhibits high mutual information with the labels and small mutual information with the data). **The remaining subspace therefore encodes a partial representation of the image without the shortcut.**

Formally, we modify the standard VAE classifier described in Section 2 as follows: we partition the latent space as $\mathbf{z} = (\mathbf{z}_1, \mathbf{z}_2)$. The classifier $C_\varphi$ uses only $\mathbf{z}_1$ as input, i.e. for a given data point $\mathbf{x}$, the predicted output is $C_\varphi\big(\mu_\phi(\mathbf{x})_1\big)$. As such, $\mathbf{z}_1$ is used for both reconstruction and classification, whereas $\mathbf{z}_2$ is used for reconstruction only.

In Figure 3, we visualize the results of this experiment in two ways: (i) Grad-CAM, where we use gradient weights both of $\mu_\phi(\mathbf{x})_1$ and of $\mu_\phi(\mathbf{x})_2$, and (ii) sampling **partial reconstructions $\tilde{\mathbf{x}}$** from the VAE: given an encoding $\mu_\phi(\mathbf{x})$, we replace each half $\mu_\phi(\mathbf{x})_i$ with standard Gaussian noise and

---

**Algorithm 1** Chroma-VAE

---

**Input:** data $\mathcal{D}_{tr}$, VAE classifier $(E_\phi, D_\theta, C_\varphi)$, $\mathbf{z}_2$-classifier $C_{\varphi'}^{(2)}$, optimization hyperparameters

//     Training initial VAE classifier

initialize $(\theta, \phi, \varphi)$

using Adam, minimize Equation (2), noting that $C_\varphi$ only takes $\mu_\phi(\mathbf{x})_1$ as input

//     Training $\mathbf{z}_2$-classifier

initialize $\varphi'$

compute the (fixed) input $\mu_\phi(\mathbf{x})_2$ using $E_\phi$

using Adam, minimize $\mathcal{L}(\varphi') = \text{CE}\Big(y, C_{\varphi'}^{(2)}\big(\mu_\phi(\mathbf{x})_2\big)\Big)$

**return** $C_{\varphi'}^{(2)}$

---

sample from the decoder:

$$\tilde{\mathbf{x}}_1 \sim D_\theta\Big( \begin{bmatrix} \mu_\phi(\mathbf{x})_1 & N(\mathbf{0}, I) \end{bmatrix}^\top \Big) \tag{3}$$

$$\tilde{\mathbf{x}}_2 \sim D_\theta\Big( \begin{bmatrix} N(\mathbf{0}, I) & \mu_\phi(\mathbf{x})_2 \end{bmatrix}^\top \Big) \tag{4}$$

We use the same predictive task and synthetic shortcut as described in Section 3.1. As can be seen from Figures 3b-c, Grad-CAM shows that $\mu_\phi(\mathbf{x})_1$ is most sensitive to pixels around the blue patch, whereas $\mu_\phi(\mathbf{x})_2$ is sensitive to pixels spread out in other regions of the image (including certain regions of the celebrity's blonde hair). The representation of the patch is largely isolated in $\mathbf{z}_1$.

Partial reconstructions in Figures 3d-e support this observation. In Figure 3d, where $\mu_\phi(\mathbf{x})_1$ is fixed and we sample only $\mathbf{z}_2 \sim N(\mathbf{0}, I)$, the decoded samples $\tilde{\mathbf{x}}_1$ represent a variety of faces that do not resemble the original image. However, *all* samples contain the shortcut patch in the bottom-right corner. Conversely, in Figure 3e where $\mu_\phi(\mathbf{x})_2$ is fixed, the reconstructed samples $\tilde{\mathbf{x}}_2$ greatly resemble the original image, but may or may not contain the shortcut. In other words, $\mu_\phi(\mathbf{x})_1$ is strongly correlated with the shortcut's presence, whereas $\mu_\phi(\mathbf{x})_2$ is *uncorrelated* with the shortcut.

Together, Grad-CAM and the partial reconstructions suggest that the latent representation of the shortcut patch has largely been sequestered into $\mathbf{z}_1$. This result is intuitive — as the classification loss on $C_\varphi$ only back-propagates through $\mathbf{z}_1$, it will learn representations that are most useful for prediction. This representation is dominated by the shortcut patch, as we showed in Section 3.1. Since the model is also simultaneously learning to reconstruct the image, most other information describing the image is partitioned into $\mathbf{z}_2$, resulting in $\tilde{\mathbf{x}}_2$ samples being very similar to the original.

### 3.3 Chroma-VAE

These results suggest a simple approach: we only need to train a separate, secondary classifier $C_{\varphi'}^{(2)}$ on $\mathbf{z}_2$ *after* the initial VAE classifier $(E_\phi, D_\theta, C_\varphi)$ has been trained, i.e. $\mathbf{z}_2$ is a fixed input into $C_{\varphi'}^{(2)}$ that is not back-propagated through to $\mathbf{x}$. Section 3.2 shows that the initial VAE classifier learns $\mathbf{z}_2$ in a way that minimizes shortcut representation but contains other salient features (by virtue of $\mathbf{z}_2$ learning to reconstruct). As such, $C_{\varphi'}^{(2)}$ will be shortcut-invariant while remaining predictive.

We name this approach **Chroma-VAE**. We provide a full description of the training procedure in Algorithm 1 and a corresponding diagram in Appendix A. From here on, we will refer to $C_\varphi$ as the $\mathbf{z}_1$-classifier ($\mathbf{z}_1$-`clf`; *not* used for prediction) and $C_{\varphi'}^{(2)}$ as the $\mathbf{z}_2$-classifier ($\mathbf{z}_2$-`clf`; shortcut-invariant classifier used for prediction).

Chroma-VAE enables high flexibility with respect to the $\mathbf{z}_2$-classifier. Since $\mathbf{z}_2$ is a latent vector, the $\mathbf{z}_2$-classifier need not be a deep neural net. Indeed, we will see in our experiments that simpler models like $k$-nearest neighbors can perform better on smaller datasets. Furthermore, instead of training $C_{\varphi'}^{(2)}$ on $\mathbf{z}_2$, we can generate partial reconstructions $\tilde{\mathbf{x}}_2$ and train a classifier directly in $\mathcal{X}$-space. This procedure allows us to exploit deep models since the inputs are now images. We term this latter approach the $\tilde{\mathbf{x}}_2$-classifier ($\tilde{\mathbf{x}}_2$-`clf`).

**Hyperparameters and Group Labels**   Our method has two hyperparameters: the dimensionality of $\mathbf{z}$, $\dim(\mathbf{z})$, and the partition fraction, $z_p = \frac{\dim(\mathbf{z}_1)}{\dim(\mathbf{z}_1) + \dim(\mathbf{z}_2)} \in (0, 1)$, which controls the relative sizes of $\mathbf{z}_1$ and $\mathbf{z}_2$. Compared to existing work, Chroma-VAE does not require training group labels. While validation group labels are technically necessary for hyperparameter tuning, we empirically found that it is possible to tune Chroma-VAE without needing them, for two key reasons: **(a)** In most domains, where the shortcut is known *a priori* (e.g. image background), we can generate and visually inspect partial reconstructions for hyperparameter tuning, by selecting $\dim(\mathbf{z})$ to ensure good reconstruction, and $z_p$ to confirm that the shortcut has been isolated in $\mathbf{z}_1$. **(b)** Even where the shortcut is unknown, worst-group accuracy is far less sensitive to Chroma-VAE hyperparameters than for existing methods such as [28]. This insensitivity is likely because most small values of $z_p$ typically suffice to isolate the shortcut in $\mathbf{z}_1$. Table 1 summarize the group label requirements.

| Example | Group-DRO [38] | JTT [28] | Chroma-VAE |
|---|:---:|:---:|:---:|
| Group Labels on Train Data | ✓ | ✗ | ✗ |
| Group Labels on Validation Data | ✓ | ✓ | ✓/✗ |

Table 1: Comparison of group label requirements. While Chroma-VAE can be tuned using validation worst-group accuracy, we found that partial reconstructions and average accuracy typically suffice.

## 4   Related Work

**Generative Classifiers**   Depending on how we decompose $p_\theta(\mathbf{x}, y)$, generative classifiers can be divided into two categories: (i) *class-conditional models* $p_\theta(y)p_\theta(\mathbf{x}|y)$ or (ii) *hybrid models* $p_\theta(\mathbf{x})p_\theta(y|\mathbf{x})$. The majority of previous work on deep generative classifiers [e.g. 40, 54] focus on the former category, e.g. for semi-supervised learning [24, 19] or adversarial robustness [40]. The most notable work in the latter category is Nalisnick et al. [32], which applies hybrid models to OOD detection and semi-supervised learning. Our work uses VAEs as the deep generative component, since we require explicit latent representation. We only focus on *hybrid* models as we desire a single model that learns the shortcut representation across all classes.

**Group Robustness**   Out-of-distribution (OOD) generalization is a broad area of study [e.g. 8, 14, 37], depending on what assumptions are made on the relationship between train and test distributions as well as the information known at train time. In this work, we primarily consider distribution shifts arising from the presence of shortcuts or spurious correlations, where signals predictive in the train distribution are no longer correlated to the label in the test distribution. Prior work has generally approached this from the *group robustness* perspective, where the objective is to maximize worst-group accuracy (typically the minority groups) while retaining strong average accuracy. To the best of our knowledge, Chroma-VAE is the first method to tackle shortcut learning from a supervised representation learning approach using generative classifiers.

Methods in this space can be distinguished by the assumptions they make. Some work rely on having group labels for training data [1, 38, 53, 39], for example, Sagawa et al. [38] optimize worst-group accuracy directly. However, as group annotations can be expensive to acquire, other approaches relax this requirement [28, 7, 44, 51, 33]. For example, Liu et al. [28], which we compare to, treat misclassified examples by an initial model as a proxy for minority groups; these samples are up-weighted when training the final model. However, their approach is brittle to hyperparameter choices, requiring group labels on validation data (from the test distribution) for hyperparameter tuning.

## 5   Experiments

In Section 5.1, we present results on the `ColoredMNIST` benchmark, a proof-of-concept which we use to highlight some key observations and comparisons. In Section 5.2, we apply Chroma-VAE to two large-scale benchmark datasets (`CelebA` and `MNIST-FashionMNIST Dominoes`), as well as a real-world problem involving pneumonia prediction using chest X-ray scans. Appendix C.1 contains further results on the `CelebA` synthetic patch proof-of-concept that was presented earlier in Section 3.

| Method | $\mathcal{D}_{in}$ | $\mathcal{D}_{out}$ |
|---|---|---|
| Theoretical UB | 75 | 75 |
| Invariant | 60.8 | 65.6 |
| Naive-Class | 87.8 | 17.8 |
| Naive-VAE-Class | 89.0 | 13.2 |
| Naive-Independent | 89.7 | 11.4 |
| JTT | 63.2 | 63.8 |
| Chroma-VAE ($\mathtt{z_1}$-clf) | 89.0 | 14.5 |
| Chroma-VAE ($\mathtt{z_2}$-clf) | **72.5** | **72.4** |

Table 2: Predictive accuracies on $\mathcal{D}_{in}$ and $\mathcal{D}_{out}$. Chroma-VAE ($\mathtt{z_2}$-clf) is the best-performing classifier. Our strongest $\mathbf{z}_2$-classifier uses $k$-nearest neighbor classifier ($k = \sqrt{N}$) rather than a MLP. Note that the best that a classifier relying only on digit can do is 75% since $p_d = 0.75$ by construction.

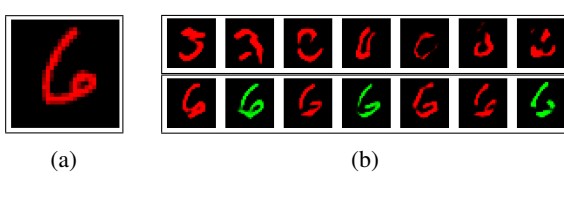

(a)  (b)

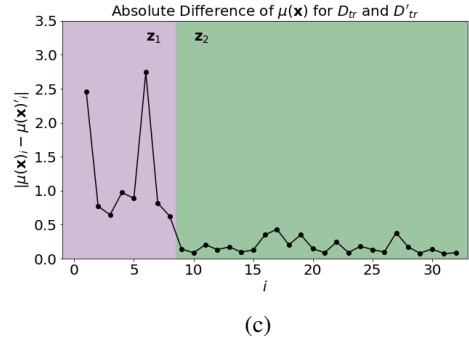

(c)

Figure 4: **(a)** Original image. **(b)** Partial reconstructions $\tilde{\mathbf{x}}_1$ *(top row)* and $\tilde{\mathbf{x}}_2$ *(bottom row)*. All samples of $\tilde{\mathbf{x}}_1$ preserve the color of the original image but not the digit. Conversely, samples of $\tilde{\mathbf{x}}_2$ preserve digit but can be variably green or red. **(c)** Plot of absolute differences between $\mu(\mathbf{x})$ of $\mathcal{D}_{tr}$ and $\mathcal{D}'_{tr}$ (the latent means) against dimension. Dimensions of $\mathbf{z}_2$ have smaller differences than $\mathbf{z}_1$.

Detailed experimental setups can be found in Appendix B. In considering baselines, we avoid comparing to methods that rely on group labels [38, 36], multiple training environments [1], or counterfactual examples [46]. We select the following baselines:

1. **Naive discriminative classifier** (Naive-Class): standard ERM classifier
2. **Naive VAE classifier** (Naive-VAE-Class): standard VAE classifier
3. **Naive VAE + classifier** (Naive-Independent): standard VAE is first trained on unlabelled data $\{\mathbf{x}_i\}_{i=1}^N$, the classifier is then separately trained on the latent projection $\{\mu_\phi(\mathbf{x}), y\}_{i=1}^N$
4. **Just Train Twice** (JTT) [28]: classifier is trained with a limited number of epochs, misclassified training points are upweighted and a second classifier is trained again

### 5.1 ColoredMNIST

**Setup.** Following Arjovsky et al. [1], (i) first we binarize MNIST [26] labels as $\hat{y}$ (with digits 0-4 and 5-9 as the two classes), (ii) we then obtain *actual* labels $y$ (used to train and evaluate) by flipping $\hat{y}$ with probability $p_d = 0.25$, and finally (iii) we obtain color labels $c$ by flipping $y$ with variable probability $p_c$. $c$ is used to color each image green or red. In the training distribution $\mathcal{D}_{in}$, $p_c = 0.1$, hence $c$ is more strongly correlated to $y$ than $\hat{y}$ is, and color becomes the shortcut. In the adversarial OOD test distribution $\mathcal{D}_{out}$, $p_c = 0.9$, i.e. every digit is more likely to be shaded the other color. Table 2 shows that the "invariant" classifier (trained on *black-and-white* images) performs similarly on both $\mathcal{D}_{in}$ and $\mathcal{D}_{out}$, hence degraded performance on $\mathcal{D}_{out}$ is solely due to shortcut learning.

The main takeaway from our results in Table 2 is that **Chroma-VAE vastly outperforms all other methods under the adversarial distribution $\mathcal{D}_{out}$**. We want to demonstrate that the $\mathbf{z}_2$-classifier is learning from a largely shortcut-invariant representation of the data in attaining this performance. First, we observe that the shortcut is heavily present in $\mathbf{z}_1$, as the $\mathbf{z}_1$-classifier performs just as *poorly* as the naive ERM approaches. Furthermore, to show that the shortcut representation is *isolated* in $\mathbf{z}_1$, we can sample and inspect partial reconstructions. We plot one input example in Figure 4b; we provide further examples in Appendix C.2. We observe that samples of $\tilde{\mathbf{x}}_1$ are color-invariant whereas samples of $\tilde{\mathbf{x}}_2$ are digit-invariant, suggesting that the bulk of color representation has been isolated in $\mathbf{z}_1$. As yet further evidence, we create $\mathcal{D}'_{tr}$, a dataset that is identical to the training set $\mathcal{D}_{tr}$ except that every image has its color flipped. We then measure $|\mu(\mathbf{x})_{(i)} - \mu(\mathbf{x})'_{(i)}|$ for every dimension $i$ of

|  | $z_p = 1/4$ | $z_p = 1/2$ | $z_p = 3/4$ |
|---|---|---|---|
| $\dim(\mathbf{z}) = 4$ |  |  |  |
| $\dim(\mathbf{z}) = 8$ |  |  |  |
| $\dim(\mathbf{z}) = 16$ |  |  |  |
| $\dim(\mathbf{z}) = 32$ |  |  |  |

Table 3: Partial reconstructions for varying choices of $\dim(\mathbf{z})$ and $z_p$. In each cell, the top row contains $\tilde{\mathbf{x}}_1$ samples and the bottom row contains $\tilde{\mathbf{x}}_2$ samples. The original image is a green "6". Increasing $z_p$ relaxes the bottleneck such that $\tilde{\mathbf{x}}_2$ samples less resemble the original image, and increasing $\dim(\mathbf{z})$ allows more information for higher fidelity reconstructions. The optimal hyperparameters that produced our results in Table 2 are $\dim(\mathbf{z}) = 32$ and $z_p = \frac{1}{4}$.

|  | $\mathcal{D}_{in}$ | $\mathcal{D}_{out}$ |
|---|---|---|
| $\beta = 100$ | 50.6 | 49.4 |
| $\beta = 10$ | 63.1 | 62.8 |
| $\beta = 5$ | 69.8 | 70.3 |
| $\beta = 1$ | 72.5 | 72.4 |
| $\beta = 0.5$ | 72.9 | 71.2 |

Table 4: Test accuracies of the $\mathbf{z}_2$-classifier on the same `ColoredMNIST` setup, where Chroma-VAE is trained using different values of $\beta$ in the ELBO objective ($\beta = 1$ recovers standard Chroma-VAE). Choosing $\beta > 1$ leads to degraded performance, worsening with higher values of $\beta$. Choosing $\beta < 1$ does not have significant impact on the test accuracies.

the latent mean, averaged across all inputs $\mathbf{x}$ in the dataset. In Figure 4c, observe that the dimensions corresponding to $\mathbf{z}_2$ have smaller differences than dimensions of $\mathbf{z}_1$, suggesting color (shortcut) is minimally contained in $\mathbf{z}_2$.

Next, the poor performance of the naive baselines highlight our finding that **generative models alone are insufficient for avoiding shortcuts.** Neither the VAE classifier (Naive-VAE-Class) nor the independent VAE + classifier (Naive-Independent) improved on the ERM model (Naive-Class). While our work motivated VAEs specifically from observations about the information bottleneck, we note that the community at-large has hoped [10] that *simply* having a generative component might incentivize the model to learn a comprehensive representation of the data-generating factors, and not just the minimum compression necessary for small training loss. Our results show that this is not true. This failure is not limited to hybrid models — further results in Section C.1 show that class-conditional generative classifiers fare just as poorly.

**Ablations for hyperparameters** $\dim(\mathbf{z})$ **and** $z_p$**.** Table 3 shows partial reconstructions as a function of varying $\dim(\mathbf{z})$ and $z_p$. As we expect, reconstructions are poor when $\dim(\mathbf{z})$ is small but improve as latent capacity increases. For sufficiently large $\dim(\mathbf{z})$, smaller values of $z_p$ are successful at isolating the shortcut color representation in $\mathbf{z}_1$, ensuring that samples of $\tilde{\mathbf{x}}_2$ are color-invariant but retain the digit shape. As $z_p$ increases, $\mathbf{z}_1$ learns both color and digit, resulting in samples of $\tilde{\mathbf{x}}_1$ that closely resemble the original image. As such, we will expect the $\mathbf{z}_2$-classifier to perform poorly since $\mathbf{z}_2$ no longer contains meaningful representations for prediction. These results suggest that hyperparameters should be tuned by first tuning $\dim(\mathbf{z})$ to ensure sufficient capacity for reconstruction,

| Method | CelebA | | | | MF-Dominoes | | Chest X-Ray | |
|---|---|---|---|---|---|---|---|---|
| | Blond/Gender | | Attractive/Smiling | | | | | |
| | Ave | Worst | Ave | Worst | Ave | Worst | Ave | Worst |
| Naive-Class | 64.2 | 29.4 | 62.5 | 21.5 | 50.4 | 0.61 | **85.0** | 10.3 |
| JTT | 65.3 | 28.3 | 64.7 | 30.0 | 50.6 | 0.81 | 64.2 | 52.3 |
| Chroma-VAE | **82.0** | **54.4** | **66.9** | **53.1** | **78.5** | **73.8** | 59.9 | **57.8** |

Table 5: Average and worst-group test accuracy for the various large-scale benchmark tasks. Chroma-VAE beats all baselines on worst-group accuracies, demonstrating that it is effective at mitigating shortcut learning at scale.

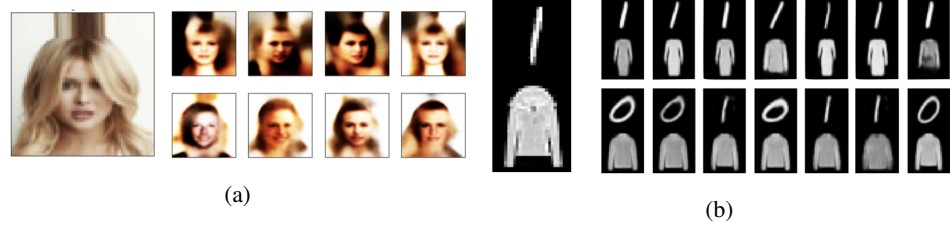

(a)

(b)

Figure 5: Partial reconstruction samples for the large-scale enchmark datasets. **(a)** `CelebA` example. The original example is **Attractive** but not **Smiling**. All samples of $\tilde{x}_1$ *(top right)* are not smiling, whereas samples of $\tilde{x}_2$ *(bottom right)* may or may not be smiling, suggesting that $z_2$ learns "smile-invariance". **(b)** `MNIST-FashionMNIST` example. All samples of $\tilde{x}_1$ *(top right)* contain the shortcut ("1" digit), whereas samples of $\tilde{x}_2$ *(bottom right)* retain the core feature (shirt image) but can have variable digit.

before tuning $z_p$ to ensure that the shortcut is learnt by $z_1$, but not the true feature. As we observe in our experiments, small values of $z_p$ typically work well.

**Ablations for $\beta$.** Our method relates to a thread of research aimed at learning (unsupervised) disentangled latent representations using VAEs. The main intuition behind these methods is enforcing independence between dimensions of the marginal distribution $q_\phi(z) := \mathbb{E}_{x \sim p_{D_{tr}}}[q_\phi(z|x)]$. One such approach is $\beta$-VAE [15], which sets the hyperparameter $\beta > 1$ (the coefficient on the Kullback-Liebler (KL) divergence term in (1)) to encourage this independence. Table 4 shows the ablation where we train Chroma-VAE with different values of $\beta$. Counterintuitively, increasing $\beta > 1$ results in degraded performance on $\mathcal{D}_{out}$. We postulate that unlike the original $\beta$-VAE, the VAE here is trained jointly with *supervision*. As such, the latent factors in the data (color and digit) are both highly correlated with the label, and therefore *with each other*, so they cannot be disentangled by $\beta$-VAE.

## 5.2 Large-Scale Benchmarks

**Setup.** We consider two benchmark `CelebA` tasks: (1) predicting **Blond** where **Gender** is the spurious feature [38, 28], and (ii) predicting **Attractive** where **Smiling** is the spurious feature [27]. We also consider the `MF-Dominoes` dataset [35], where input images consist of `MNIST` digits (0 or 1) concatenated with `FashionMNIST` objects (coat or dress), with the FashionMNIST object being the true label. For both of these datasets, we use the *harder* setting from Lee et al. [27], where the spurious feature is *completely correlated* with the label at train time (by filtering out the two minority groups). Chroma-VAE is naturally suited for this regime, as complete correlation allows the spurious feature to be simultaneously compressible and highly discriminative.

In addition, as an example of a real-life domain where shortcut learning happens, we consider the prediction of pneumonia from frontal chest X-ray scans [52, 36, 21]. The shortcut is the background of the image, which contains artifacts specific to the machine that took the X-ray. As different hospitals use different machines and have differing rates of diagnosis, the background becomes spuriously correlated with the label. There is no benchmark dataset for this problem and different

authors have created their own datasets from existing public or private X-ray repositories. Likewise, we make a training set ($N = 20$K) with roughly equal numbers of X-rays from the National Institutes of Health Clinical Center (NIH) dataset [48] and the Chexpert dataset [18], such that 90% of NIH images are from the negative class and 90% of Chexpert images are from the positive class.

Table 5 summarizes the results. On `CelebA` and `MF-Dominoes`, **Chroma-VAE has the highest average and worst-group accuracies.** Figure 5 shows the corresponding partial reconstructions, which visually confirm that the desired invariances have been learnt and the shortcuts correctly isolated. JTT underperforms in this complete correlation setting (no minority groups) as it is unable to leverage misclassification of minority examples to train the final model. This experiment highlights a significant difference between group robustness methods and Chroma-VAE. Methods like JTT perform poorly when there are fewer minority examples, as they rely on these examples to approximate the test distribution and implicitly teach the model what features are most useful. In contrast, Chroma-VAE learns *explicit* representations of the shortcut and true features. Chroma-VAE performs *better* when the correlation between the shortcut and label is stronger, as the shortcut is then more likely to be the minimal encoding that is predictive of the label.

On the Chest X-ray dataset, **Chroma-VAE has the highest worst-group accuracy**; however, we note that it suffers from lower average accuracy. The fact that Chroma-VAE improves on worst-group accuracy shows that the model works correctly to avoid shortcut learning by isolating the shortcut in $\mathbf{z}_1$. However, the trade-off against average accuracy also highlights one limitation of our approach, which is the reliance on VAE architecture to model the data distribution well. To the best of our knowledge, we are the first to attempt to model Chest X-ray data with a deep generative model, as existing work for Chest X-ray shortcut learning rely on discriminative models — specifically, pre-trained ResNets [e.g. 36]. As the VAE is unlikely to model the X-ray dataset perfectly, some meaningful predictive features will not be well-captured, resulting in poorer performance of the $\mathbf{z}_2$-classifier despite being shortcut-free. Reconstruction examples for the Chest X-ray dataset can be found in Appendix C.3.

# 6   Discussion

We empirically observe that shortcuts — being the most efficient and predictive compression of the data — are often preferentially encoded by deep neural networks. Inspired by this result, we propose Chroma-VAE, which exploits a VAE classifier to learn a latent representation of the image where the shortcut is abstracted away. This representation can be used to train a classifier that generalizes well to shortcut-free distributions. We demonstrate the efficacy of Chroma-VAE on several benchmark shortcut learning tasks.

A limitation of our work is the reliance on VAEs to model the underlying data distribution well, which can be challenging for many natural image datasets. Extending Chroma-VAE to incorporate stronger deep generative models, such as diffusion-based models, could lead to stronger performance on many real-life datasets. Indeed, in principle our pipeline is not anchored to the VAE, and could be used with other families of deep generative models.

Moreover, many spurious correlations may not be easily described with low information. For example, the spurious feature can be the entire background (such as water or land in the `Waterbirds` dataset [38]), which can contain a relatively large amount of information. In the future, it would be exciting to generalize Chroma-VAE to learn richer latent representations that compartmentalize different types of features in the inputs, rather than simply segmenting low-information shortcuts from the rest of the image. This outcome could possibly be achieved by introducing priors that explicitly encourage different subspaces of $\mathbf{z}$ to correspond to different features.

**Acknowledgements**

We would like to thank Marc Finzi, Pavel Izmailov, and Wesley Maddox for helpful comments. This research is supported by NSF CAREER IIS-2145492, NSF I-DISRE 193471, NIH R01DA048764-01A1, NSF IIS-1910266, NSF 1922658 NRT-HDR: FUTURE Foundations, Translation, and Responsibility for Data Science, Meta Core Data Science, Google AI Research, BigHat Biosciences, Capital One, and an Amazon Research Award.

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
