**Outline of Appendices** Appendix A describes Chroma-VAE's model and training procedure in detail. Appendix B describes the experimental details in Sections 3 and 5. Appendix C contains further results referenced in the paper. Appendix D is a statement on the societal impact of our work.

# A  Chroma-VAE: Model Implementation

Figure 6 depicts Chroma-VAE and its training procedure. In the diagrams below, the forward-facing solid arrows represent the forward pass and the backwards-facing dotted arrows represent the backpropagation of gradients.

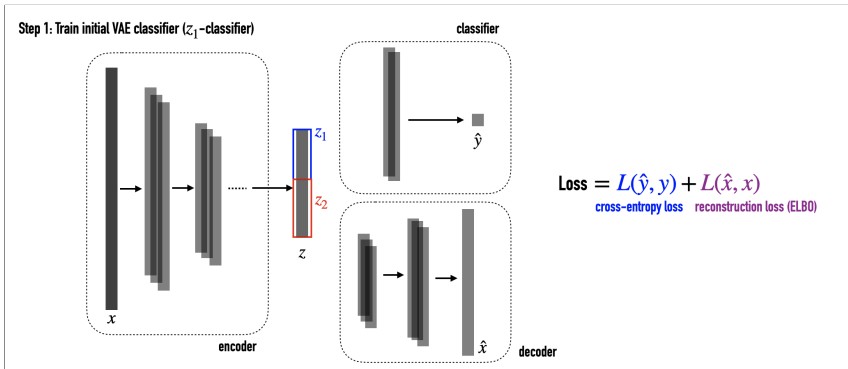

(a) In the first stage of the training procedure, the encoder, decoder, and classifier are all trained. The loss function consists of two terms: the cross-entropy loss and the reconstruction loss.

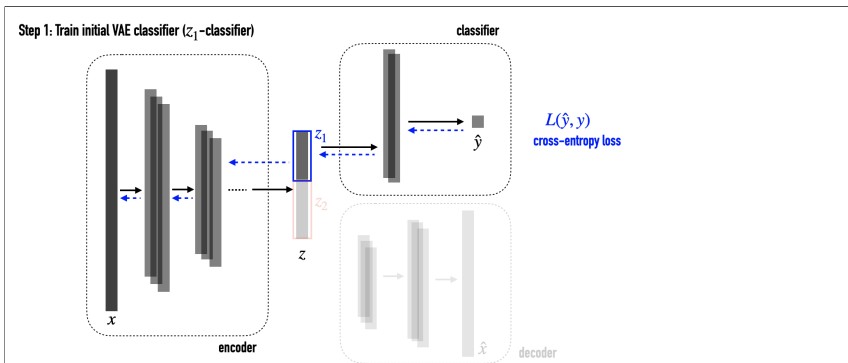

(b) The classifier head is attached to $\mathbf{z}_1$ only (*not* the entire latent embedding $\mathbf{z}$). As such, the cross-entropy loss term is only backpropagated through $\mathbf{z}_1$.

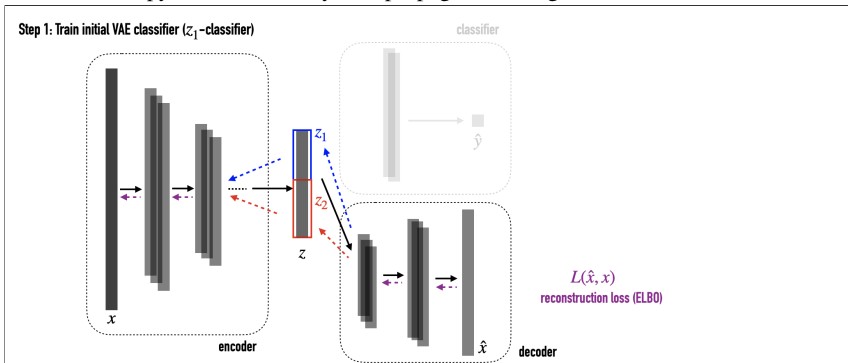

(c) The decoder is attached to the entire latent embedding $\mathbf{z}$. As such, the reconstruction loss term is backpropagated through both $\mathbf{z}_1$ and $\mathbf{z}_2$. Consequently, putting both terms of the loss function together, $\mathbf{z}_1$ learns to encode shortcut features whereas $\mathbf{z}_2$ encodes salient non-shortcut features of the input image.

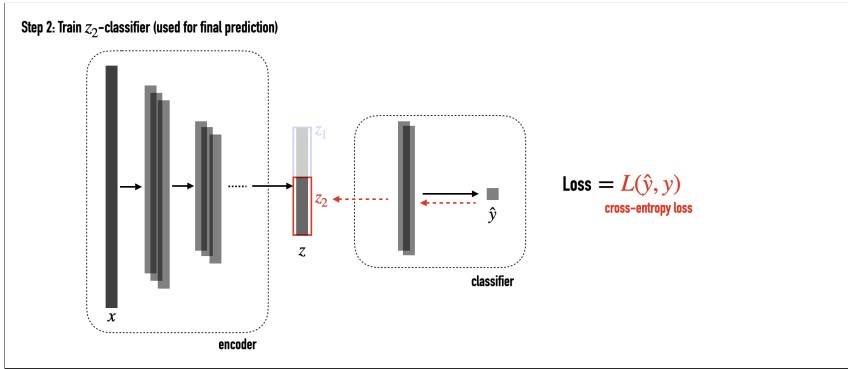

(d) In the second stage of the training procedure, we discard the original classifier head (attached to $\mathbf{z}_1$) and attach a new one to $\mathbf{z}_2$ only instead. We train this classifier using the same cross-entropy loss, except that gradients are now being backpropagated to $\mathbf{z}_2$. Note that the encoder and decoder are *fixed* at this stage, and so $\mathbf{z}$ itself is fixed. Only the parameters of the new classifier head are being updated. Since $\mathbf{z}_2$ was trained to encode non-shortcut representations, the classifier learns to predict the label using non-shortcut features only.

Figure 6: Diagram of Chroma-VAE's training procedure, showing loss terms and backpropagated gradients.

# B    Experimental Details

**Section 3: CelebA (Synthetic Patch)**    We use standard CNN architecture. The encoder contains 4 convolutional layers, followed by fully-connected layers to $\mu_\phi$ and $\sigma_\phi^2$. The decoder contains a fully-connected layer, followed by 4 deconvolutional layers. The Adam optimizer [22] is used with a learning rate of 0.0005. The hyperparameters are $\dim(z) = 128$ and $z_p = 0.5$.

**Section 5: ColouredMNIST**    We use standard CNN architecture. The encoder contains 2 convolutional layers, followed by fully-connected layers to $\mu_\phi$ and $\sigma_\phi^2$. The decoder contains a fully-connected layer, followed by 2 deconvolutional layers. The Adam optimizer is used with a learning rate of 0.001. The hyperparameters are $\dim(z) = 32$ and $z_p = 0.25$. We performed a hyperparameter sweep for JTT with $T \in \{1, 3, 5, 10\}$ and $\alpha \in \{2, 5, 50, 100\}$.

**Section 5: CelebA and MF-Dominoes**    We use standard CNN architecture. The encoder contains 4 convolutional layers, followed by fully-connected layers to $\mu_\phi$ and $\sigma_\phi^2$. The decoder contains a fully-connected layer, followed by 4 deconvolutional layers. The Adam optimizer is used with a learning rate of 0.0005. For CelebA, the hyperparameters are $\dim(z) = 128$ and $z_p = 0.5$. For MF-Dominoes, the hyperparameters are $\dim(z) = 8$ and $z_p = 0.5$. We performed a hyperparameter sweep for JTT with $T \in \{1, 3, 5, 10\}$ and $\alpha \in \{2, 5, 50, 100\}$.

**Section 5: Chest X-rays**    Following prior work [52], we use a Densenet (pretrained on ImageNet) [17] for the encoder, along with a standard CNN decoder (containing a fully-connected layer, followed by 4 deconvolutional layers). The Adam optimizer is used with a learning rate of 0.0001. The hyperparameters are $\dim(z) = 128$ and $z_p = 0.5$.

# C  Additional Results

| Method | $\mathcal{D}_{tr}$ | $\mathcal{D}_{neut}$ | $\mathcal{D}_{anti}$ |
|---|---|---|---|
| Invariant | 95.58 | 94.57 | 92.07 |
| Naive-Class | **99.83** | 90.90 | 4.45 |
| Naive-VAE-Class | 99.65 | 89.47 | 4.69 |
| Naive-Independent | 99.64 | 91.49 | 9.86 |
| Class-Conditional | 93.52 | 61.19 | 3.95 |
| JTT | 99.62 | **92.76** | 6.52 |
| Chroma-VAE ($\mathbf{z}_1$-clf) | 99.64 | 92.33 | 6.03 |
| Chroma-VAE ($\mathbf{z}_2$-clf) | 95.57 | 91.82 | **87.24** |

Table 6: Predictive accuracy on the various test distributions. On $\mathcal{D}_{anti}$, Chroma-VAE improves on the naive discriminative classifier by a factor of 20.

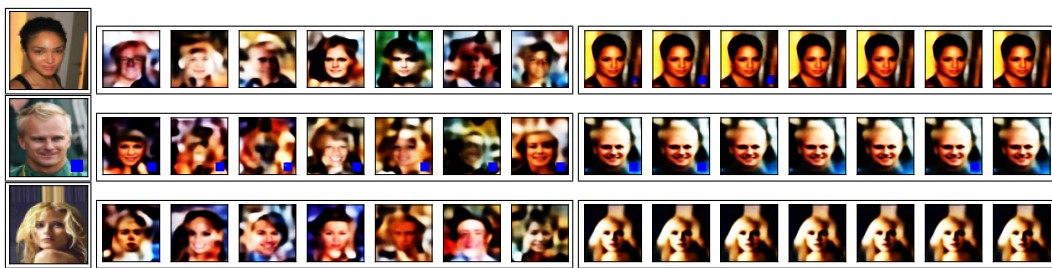

Figure 7: **For each row, left to right: (a)** original image, **(b)** samples of partial reconstructions $\tilde{\mathbf{x}}_1$, **(c)** samples of partial reconstructions $\tilde{\mathbf{x}}_2$.

## C.1  Synthetic Patch on CelebA

We consider the same task as in Section 3.1, where we predict blond hair, with a synthetic shortcut (blue patch superimposed onto the positive class with probability 0.9). We evaluate performance on three test environments:

- **Training Distribution** ($\mathcal{D}_{tr}$): the patch is added to all members of the positive class
- **Neutral Distribution** ($\mathcal{D}_{neut}$): no patches are added
- **Adversarial Distribution** ($\mathcal{D}_{anti}$): the patch is added to all members of the *negative* class

Table 6 summarizes the results. **Chroma-VAE vastly outperforms all other methods under the adversarial distribution**. The $\mathbf{z}_2$-classifier has the same performance as the invariant classifier on the in-distribution $\mathcal{D}_{tr}$, implying that it did not benefit (unfairly) from shortcut representations, unlike the naive baselines. However, the fact that the $\mathbf{z}_2$-classifier is still outperformed by the invariant classifier on $\mathcal{D}_{anti}$ and $\mathcal{D}_{neut}$ suggests that not all desired features were captured by $\mathbf{z}_2$. This implies that the maximally-useful compression of the data in $\mathbf{z}_1$ primarily encodes the shortcut, but also other useful correlations.

Partial reconstructions in Figure 7 support the quantitative results. The first image represents an example of the negative class (non-blonde hair), the second image is an example of the positive class with the shortcut, and the third image is an example of the positive class *without* the shortcut (which happens with probability 0.1 in the training set). Similar to Figure 3, samples of $\tilde{\mathbf{x}}_1$ do not resemble the original image, but always encode the shortcut patch if it exists in the original image. Conversely, samples of $\tilde{\mathbf{x}}_2$ bear great resemblance to the original image but may or may contain the shortcut — confirming that they are shortcut-invariant.

## C.2 ColoredMNIST: Partial Reconstructions

Figure 8 shows partial reconstruction samples on three additional images. We observe the same patterns as in Figure 4, where samples of $\tilde{x}_1$ preserve the colour of the original image but not the digit, whereas samples of $\tilde{x}_2$ preserve the digit of the original shape but is colour-invariant.

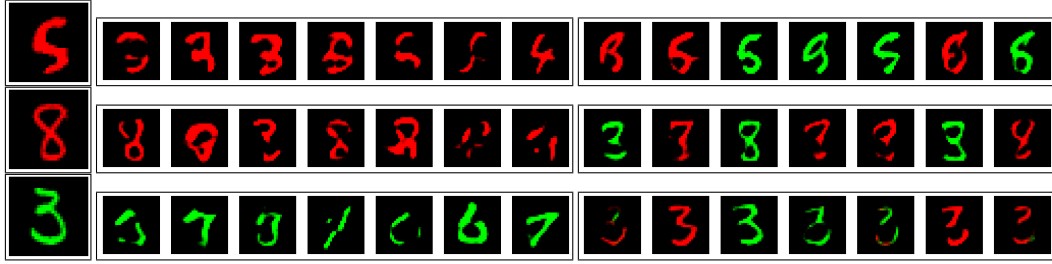

Figure 8: **For each row, left to right: (a)** original image, **(b)** samples of partial reconstructions $\tilde{x}_1$, **(c)** samples of partial reconstructions $\tilde{x}_2$.

## C.3 Chest X-Ray: Partial Reconstructions

Figure 9 shows the reconstruction as well as partial reconstruction samples on a test example. Similar to partial reconstructions on the CelebA dataset, samples of $\tilde{x}_1$ are more diverse and less likely to resemble the original image, whereas samples of $\tilde{x}_2$ bear greater resemblance to the original image, especially around the center region of the image (where true predictive signals for pneumonia are located).

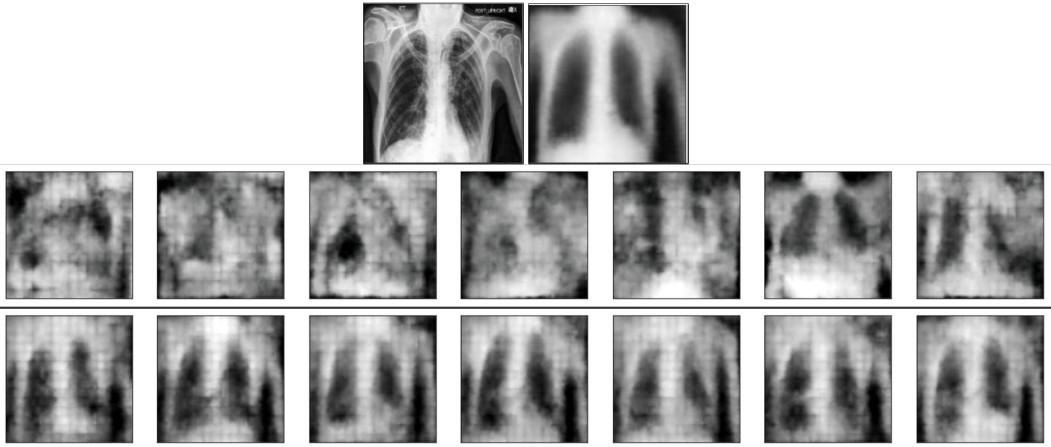

Figure 9: **Top to bottom, left to right: (a)** original image, **(b)** reconstruction, **(c)** samples of partial reconstructions $\tilde{x}_1$, **(d)** samples of partial reconstructions $\tilde{x}_2$.

## D  Broader Societal Impact

**Positive Sources of Impact**   By learning shortcut-invariant predictive signals, Chroma-VAE is robust to distribution shifts and performs well when deployed on OOD test cases. This is promising for applications in high-stakes domains where errors are costly, such as the pneumonia prediction example in Section 5.2. Furthermore, the partial reconstructions produced by Chroma-VAE can be useful for identifying if shortcut learning is occurring in the first place, as they can be examined to identify traits that are invariant in each latent subspace.

**Potential for Abuse**   Deep generative models can be used to produce *deepfakes* [49], which are samples generated by the model that are realistic enough to resemble real-life examples, e.g. fake faces produced by a generative model trained on `CelebA`. Chroma-VAE can be abused in a similar (and potentially more dangerous) way, as partial reconstructions can be generated to keep certain traits while changing others, by concatenating $z_1$ and $z_2$ from different samples. For example, partial reconstructions produced by Chroma-VAE trained on gender labels might be used to generate "counterfactual deepfakes", e.g. a person with the opposite gender. Combating deepfakes is an active area of research, and such techniques can be used to detect Chroma-VAE deepfakes as well.