# OpenReview forum: "Chroma-VAE: Mitigating Shortcut Learning with Generative Classifiers"
_NeurIPS.cc/2022/Conference — NeurIPS 2022 Accept_

### Official Review · Reviewer_ef3p · 2022-07-07

**Rating:** 4
**Confidence:** 3
**Soundness:** 2 fair
**Presentation:** 4 excellent
**Contribution:** 3 good

**Summary:**

This paper addresses the problem of shortcut learning in deep neural networks. First, the paper makes the observation that shortcut features tend to be the ones that are preferentially encoded when a limited amount of information can be encoded in one of the hidden layers of the network. The paper then builds on this observation to propose a two-stage approach for learning a classifier robust to shortcut solutions: first a VAE classifier is learned whereby the classification is only based on a small part of the latent, second an independent classifier is trained on the part of the latent space of the VAE that did not feed into the classifier. The efficacy of the method is demonstrated using several hard versions of ‘shortcut benchmarks’ in which the shortcut features are strongly or completely correlated with the targets.

**Questions:**

I would like to ask the authors to include evaluations of the proposed Chroma-VAE method on “softer” versions of the benchmarks in which the shortcut feature is only weakly correlated with the target and/or the authors should provide convincing evidence that the Chroma-VAE method can be useful in real-world settings (or discuss clearly why the currently reported real-world experiment has practical value).

**Limitations:**

If my main concern with this study holds (see above), the limitation that the proposed method depends on the shortcut feature being very strongly correlated with the target should be discussed.

I think the potential negative societal impacts are satisfactorily addressed.

**Strengths And Weaknesses:**

The paper is well written, insightful and a pleasure to read. I am not very familiar with shortcut learning benchmarks, but I had the feeling I was able to follow the entire paper without too much effort.

The proposed solution for overcoming shortcut learning, the Chroma-VAE, is well motivated, clearly explained and to the best of my knowledge novel and original.

The performance of the Chroma-VAE is very strong on the toy dataset benchmarks. The paper provides a clear intuition for why the method is able to perform so well, which is backed up by insightful visualizations

In general I would like to support acceptance of this paper, but I do have one important concern that should be addressed first. My concern is that I suspect that the proposed Chroma-VAE method will only work well in ‘artificial’ situations where the shortcut feature that should be avoided is very strongly correlated with the target. If I’m not mistaken, in general, the part of the VAE’s latent space connected to the classifier should encode all features relevant for the classification task (i.e., both the shortcut feature and the true signal) while the unconnected part of the latent space should encode only irrelevant features. It seems to me that only when the shortcut features are very strongly correlated with the target, some other features with relevance for the classification problem will be left to be encoded in the unconnected part of the latent space. Perhaps by restricting the size of the part of the latent space connected to the classifier it is possible in practice to encourage that mostly the shortcut feature is encoded there, but on softer / realistic benchmarks I do not think this will be straightforward and I expect it might lead to low performance.
Therefore, to investigate this, I suggest that the authors additionally include experiments on “softer” versions of the shortcut benchmarks in which the shortcut features are less strongly correlated with the target.

The authors do include one experiment on a “real-world”, X-ray dataset, but the results of the Chroma-VAE method on this experiment are not very convincing, as there is a large drop in average accuracy. I am not familiar with this benchmark, but it seems to me that this is not a result with any practical value. (If I’m mistaken, please correct me. But in what practical situation would we prefer the model with the lower average accuracy?) I therefore think that the claim that “our method is viable on a natural, real-world data set” must be moderated, or else better motivated.

Minor points:
- The result mentioned in L270-271 seems to be missing from Table 3.
- It would be good to make the caption of Table 4 clearer (e.g., to include that these are the results for the X-Rays experiment).

---

> ### Author Response · Authors · 2022-08-02
> **Response to Reviewer ef3p**
>
> First of all, we greatly appreciate your kind words on the writing quality and presentation of our paper, which we have put great care into. As you have noted in your review, Chroma-VAE is novel and effective on the benchmark experiments that we have tested it on. Both the quantitative results and qualitative visualizations support our claims that Chroma-VAE can learn shortcut-invariant representations for downstream classification tasks.
>
> The singular concern that you raised in your review is whether Chroma-VAE has value in real-world settings. We will address both questions that you raised in voicing this concern, which is (i) whether Chroma-VAE can perform well on “softer” scenarios where the shortcut is only weakly-correlated to the label, and (ii) better motivating our results on the real-life Chest X-ray dataset, where our average accuracy was lower despite better WGA.
>
> **(I) Strong vs. weak correlation**
>
> We believe the importance and value of performing effectively on the “strong correlation” setting has been understated. There are two points that we would like to bring up here.
>
> (1) First, we note that **shortcuts can be strongly correlated with the label in many salient real-life situations**, not just in “artificially-created” datasets. For example, the canonical example often cited by the community, where cows in exotic/non-common backgrounds are mis-classified, comes about precisely because cows **only** exist on common backgrounds like green fields or pastures in datasets like ImageNet or MS-COCO [1] — one simply just can’t find cows on a basketball court in real-life! Another real-life example is traffic classification for self-driving cars, where the image background (e.g. night or day) is often a shortcut for the object class (e.g. pedestrian or vehicle). Since there are far fewer pedestrians at night than during the day, the background becomes strongly correlated with the label in naturally-collected traffic datasets.
>
> (2) The presence of **this strong correlation is what causes shortcut learning to be such a challenging problem**. As mentioned in our general response, shortcuts become shortcuts precisely because the model prefers to learn them instead of the true features. This preference comes about from the combination of two factors: shortcuts are simple (favored by the inductive biases of the network) and strongly correlated (predictive of label and leads to low training loss). In situations where spurious features are only weakly correlated to the label, the network has far less incentive to learn them and is not deterred from picking up true signals.
>
> To support this point, we conducted additional experiments with varying levels of correlation, using the same MNIST-FashionMNIST dataset that we considered in our paper (which only showed the 100% correlation setting). The table below shows the results of our experiments.
>
> First consider the results of the naive classifier. By having just 5% of the minority groups present, the classifier can achieve a WGA of 74%, up from 0% in a fully-correlated setting. However, further increases in the minority group data only leads to slight increases in WGA — with a fully balanced dataset, the WGA is only 79%. As such, we can see that even weakening the shortcut-label correlation by a little encourages the model towards learning the true solution. Spurious correlation is therefore most harmful when the level of correlation is strong. It is at these settings where using Chroma-VAE can lead to the most significant improvements in performance.
>
> The Chroma-VAE results further emphasize this point. With complete correlation (0% minority groups), the z_2 classifier learns the shortcut-invariant representation and achieves a WGA of 80%. As the percentage of minority group data increases, it is instead the z_1 classifier that learns the true salient signal. This further corroborates the results of the naive classifier, showing that in this dataset, having a strong correlation is critical for the MNIST signal to be considered as a shortcut. With even a small amount of minority group data, the network prefers instead to encode the true signal.
>
> | Minority Group Percentage |  Naive Classifier |       | Chroma-VAE |       |                  |
> |---------------------------|-------------------|-------|------------|-------|------------------|
> |                           | Mean              | Worst | Mean       | Worst | Which classifier |
> | 0%                        | 0.505             | 0.008 | 0.881      | 0.808 | z_2              |
> | 5%                        | 0.892             | 0.741 | 0.891      | 0.802 | z_1              |
> | 10%                       | 0.899             | 0.769 | 0.915      | 0.822 | z_1              |
> | 20%                       | 0.897             | 0.786 | 0.920      | 0.814 | z_1              |
> | 50%                       | 0.906             | 0.794 | 0.913      | 0.815 | z_1              |

---

> > ### Comment · Reviewer_ef3p · 2022-08-03
> > **Response to author rebuttal**
> >
> > Thanks to the authors for their response to my review.
> >
> > I very much appreciate the detailed response which is very helpful to get a better understanding of the proposed method. Unfortunately, however, I’m afraid that the response has not taken away my concern that the proposed method might only perform well in artificial situations.
> >
> >
> > (I) Strong vs weak correlation
> >
> > To start with, the authors make a reasonable convincing case that in real-life situations shortcuts *can be* strongly correlated with the label. This helps the case of the paper (and I think this should be included in the paper itself as well), but it is not enough. The artificial examples considered in the paper are still too far away from any real-life problems. The example of the cows in ImageNet or MS-COCO that is brought up seems like a good opportunity to test the proposed method in a more realistic setting. Including such an experiment could considerably strengthen the paper and perhaps disprove my concern.
> >
> > Next, the authors make the argument that strong correlation is what causes shortcut learning to be such a challenging problem. They back this up by a new experiment in which they show that a naive classifier suffers a lot from shortcut learning with 100% correlation, but that when this percentage is only slightly reduced the naive classifiers suffers a lot less from shortcut learning. However, I’m afraid this only strengthens my concern that the 100% correlation setting is artificial. Moreover, unless I’m mistaken, already when the correlation is slightly reduced, the Choma-VAE starts to obtain better performance when z_1 is used as classifier rather than z_2. Isn’t this also an argument that the proposed method (which uses z_2) is specifically suited for the 100% correlation setting?
> >
> >
> > (II) Chest X-ray Dataset Results
> >
> > I do not agree that the result that the Chroma-VAE does better on WGA at the cost of average accuracy is a demonstration that the methodology is “fundamentally sound”. I’m afraid I still think that these results on the X-ray dataset are not of any practical value. The authors suggest the low performance might be due to the inherent difficulty of training the VAE architecture on the X-ray data. This might well be the case, but the VAE architecture is a critical part of the method.
> >
> > To summarize, I’m afraid that I still have the important concern that the proposed method will only work well in ‘artificial’ situations where the shortcut feature that should be avoided is very strongly correlated with the target.
> >
> > As minor comment, it seems that the minor comments from my original review have remained unaddressed.

---

> > > ### Author Response · Authors · 2022-08-09
> > > **Response to Reviewer ef3p (2)**
> > >
> > > Thank you for reading through our detailed response and for responding promptly, we definitely appreciate your time and effort. We would like to follow up on your reply:
> > >
> > > **(I) Choice of datasets being sufficient for shortcut learning literature**
> > >
> > > In your reply, you noted that we should test our method on a dataset like MS-COCO. However, we believe it is important to point out here that ImageNet and MS-COCO are not datasets that the shortcut learning or spurious correlation community use as benchmarks. Indeed, while the cow example is often cited as an example because of the seminal observation made by [1], _comprehensively_ avoiding shortcut learning in large-scale object detection and classification models with dozens of classes is a far more challenging problem that the current community has not yet tackled. Where ImageNet is considered for OOD generalization, it is generally in the context of robustness to common corruptions or perturbations (ImageNet-C [2] being the benchmark dataset in this area) rather than shortcut learning or spurious correlations.
> > >
> > > In this light, we believe that the choice of datasets that we have experimented on is sufficiently diverse and representative and meets the bar for shortcut learning literature. Indeed, all the datasets that we have considered are widely-used benchmarks — **CelebA** in spurious correlation works like [3, 4, 5, 6], **Colored MNIST** in invariant prediction works like [7, 8], and **FashionMNIST-MNIST** in simplicity bias works like [6, 9, 10]. On all of these datasets, we have demonstrated that our method can learn a shortcut-invariant representation.
> > >
> > > **(II) FashionMNIST-MNIST and the strong correlation setting**
> > >
> > > In prefacing our response with (I), we would like to draw the discussion back to the strong correlation setting.
> > >
> > > First and foremost, we respectfully reiterate the point that a strong correlation setting, even to the extent of 100% correlation, is (i) not “artificial” and (ii) is not a setting with any less value than mitigating shortcut learning with weaker correlation. In support of point (i), we highlighted several examples in our previous response showing how strong or near-perfect correlation can occur in many real-life scenarios. In support of point (ii), we note that we are not the first work to consider such a setting — there are various existing works [e.g. 5, 11] that **solely** focus on the 100% correlation setting. At the very least, mitigating shortcut learning in “strong” settings already has value in itself.
> > >
> > > Rather, the concern that you raise here seems to ultimately stem from the discrepancy between (a) us noting the many “real-life” scenarios for which strong or almost perfect correlation to shortcuts can hold, and (b) the narrower set of benchmarks that our paper experiments on. To this end, we note that shortcut learning is itself a relatively new subfield, with a small number of well-established benchmarks. To be able to compare our work to existing literature, we have selected datasets that various other papers in the field are familiar with and have also used. Our methodology also follows from existing work —  for example, the 100% correlation setting that we use in Section 5.2 of our paper is indeed the same as what [5] does. In lieu of well-established, “real-life” datasets like MS-COCO for shortcut learning specifically, controlling correlation within benchmark datasets is an accepted and effective way to test stronger correlation settings. As methods in this field develop and evolve, we similarly expect benchmark datasets used by the community to grow.
> > >
> > > Finally, the additional FashionMNIST-MNIST result that we provided in our response above is also meant to clarify the relationship between correlation and shortcuts. The key assumption that Chroma-VAE makes is that there are features in the dataset that a model preferentially encodes, especially under an information bottleneck. However, this preference is itself _a function of correlation_. This relationship depends on the dataset and the relative complexity of the shortcut feature vs. the salient feature. In FashionMNIST-MNIST, for example, the true feature is itself not very complex, as such, at lower levels of correlation, the model can easily learn the salient feature. However, in both the Colored MNIST and Chest X-ray experiments in our paper, shortcut learning persists even at weaker levels of correlation, because the shortcut is far easier to encode than the desired feature of interest.
> > >
> > > **--------------------------**
> > >
> > > We hope that our follow-up reply has provided additional clarity about both the merits of our work as well as the experimental choices that we have made.

---

> > > ### Author Response · Authors · 2022-08-09
> > > **Response to Reviewer ef3p (2, continued)**
> > >
> > > [1] Sara Beery, Grant Van Horn, and Pietro Perona. Recognition in terra incognita. 2018.
> > >
> > > [2] Dan Hendrycks and Thomas Dietterich. Benchmarking Neural Network Robustness to Common Corruptions and Perturbations. 2019.
> > >
> > > [3] Shiori Sagawa, Pang Wei Koh, Tatsunori B. Hashimoto, Percy Liang. Distributionally Robust Neural Networks for Group Shifts: On the Importance of Regularization for Worst-Case Generalization. 2019.
> > >
> > > [4] Sagawa, Percy Liang, and Chelsea Finn. Just train twice: Improving group robustness without training group information. 2021.
> > >
> > > [5] Yoonho Lee, Huaxiu Yao, and Chelsea Finn. Diversify and disambiguate: Learning from underspecified data. 2022.
> > >
> > > [6] Polina Kirichenko, Pavel Izmailov, and Andrew Gordon Wilson. Last layer re-training is 410 sufficient for robustness to spurious correlations. 2022.
> > >
> > > [7] Martin Arjovsky, Léon Bottou, Ishaan Gulrajani, and David Lopez-Paz. Invariant risk minimization. 2019.
> > >
> > > [8] David Krueger, Ethan Caballero, Joern-Henrik Jacobsen, Amy Zhang, Jonathan Binas, Dinghuai Zhang, Remi Le Priol, Aaron Courville. Out-of-Distribution Generalization via Risk Extrapolation (REx). 2020.
> > >
> > > [9] Damien Teney, Ehsan Abbasnejad, Simon Lucey, Anton van den Hengel. Evading the Simplicity Bias: Training a Diverse Set of Models Discovers Solutions With Superior OOD Generalization. 2022.
> > >
> > > [10] Matteo Pagliardini, Martin Jaggi, François Fleuret, Sai Praneeth Karimireddy. Agree to Disagree: Diversity through Disagreement for Better Transferability. 2022.
> > >
> > > [11] Alexander D'Amour, Katherine Heller, Dan Moldovan, Ben Adlam, Babak Alipanahi, Alex Beutel, Christina Chen, Jonathan Deaton, Jacob Eisenstein, Matthew D. Hoffman, Farhad Hormozdiari, Neil Houlsby, Shaobo Hou, Ghassen Jerfel, Alan Karthikesalingam, Mario Lucic, Yian Ma, Cory McLean, Diana Mincu, Akinori Mitani, Andrea Montanari, Zachary Nado, Vivek Natarajan, Christopher Nielson, Thomas F. Osborne, Rajiv Raman, Kim Ramasamy, Rory Sayres, Jessica Schrouff, Martin Seneviratne, Shannon Sequeira, Harini Suresh, Victor Veitch, Max Vladymyrov, Xuezhi Wang, Kellie Webster, Steve Yadlowsky, Taedong Yun, Xiaohua Zhai, D. Sculley. Underspecification Presents Challenges for Credibility in Modern Machine Learning. 2020.

---

> ### Author Response · Authors · 2022-08-02
> **Response to Reviewer ef3p (continued)**
>
> **(II) Chest X-ray Dataset Results**
>
> We also want to address our results on the Chest X-ray dataset and show that despite the lower average accuracy, our findings are still promising for the validity of our approach.
>
> In general, X-ray data is difficult to model and reconstruct. Existing classifiers on CXR datasets use ImageNet pre-trained models entirely [e.g. 2],  and to the best of our knowledge, there is no work out there that use VAEs to generatively model CXR data. Since our method is VAE-based and requires the model to reconstruct well, we believe that the inherent difficulty of training the VAE architecture is responsible for the poorer average accuracy, not a failing in the principle of our method. For example, we had to use a pre-trained ImageNet encoder paired with a deconvolutional decoder. (We found that a CNN-based encoder fared even worse.)
> Our primary goal for this experiment is not to be competitive with state-of-the-art, but to show that the principle of our approach works on real-life data, not merely “synthetic” and easily-modeled datasets like MNIST. Indeed, despite the poorer average accuracy, our method still beat the other baselines in terms of WGA. Indeed, this shows that the methodology behind Chroma-VAE is fundamentally sound and does successfully mitigate the network from learning shortcuts.
>
> We believe that Chroma-VAE has immense value in real-world settings where shortcut learning occurs and we hope that our response has alleviated some of your concerns on this front. We thank you again for your supportive and kind words, and would greatly appreciate you raising the score if you are satisfied by our response.
>
>
> [1] Sara Beery, Grant Van Horn, and Pietro Perona. Recognition in terra incognita. 2018.
>
> [2] Aahlad Puli, Lily H. Zhang, Eric K. Oermann, Rajesh Ranganath. Out-of-distribution Generalization in the Presence of Nuisance-Induced Spurious Correlations. 2022.

---

### Official Review · Reviewer_aZBR · 2022-07-11

**Rating:** 7
**Confidence:** 3
**Soundness:** 3 good
**Presentation:** 3 good
**Contribution:** 3 good

**Summary:**

The authors presented a method for mitigating shortcuts using generative classifiers. The main idea is to isolate the latent factors corresponding to the shortcut in the VAE latent space and use complementary space for learning the classifier. The underlying intuition is that the shortcuts are easy to compress features, so they will only be present/detectable in a small fraction of the latent space embedding. Experiments show the efficacy of the method on multiple datasets.

**Questions:**

1. It is not clear why gender would be a shortcut and not the hair color, if both are correlated with the output labels.
2. In Figure 4. C, the dimensions of z_1 is more than z_2. Shouldn’t it be smaller since z_1 is correlated to shortcuts and thus more compressible?
3. What does ‘invariant’ means in  Table2.
4. Why do you think KNN gives better accuracy?
5. “MLPs with smaller hidden layers have better accuracies, and our top performance is a k-nearest neighbor classifier with k = N^(½).” What does N denote here?


**Limitations:**

Yes, the authors have addressed the limitations and potential negative societal impact in the appendix.

**Strengths And Weaknesses:**

### **Strengths**

1. The paper is well written and easy to follow.
2. The idea is well-motivated and explained using information bottleneck theory and GradCam. The method separates the shortcut in the latent space of VAE and uses the remaining space to train the classifier invariant to the shortcuts.
3. The paper is trying to solve an important problem of mitigation shortcuts and spurious correlations.
4. The authors presented intensive experiments with detailed analysis. The methods show clear performance gains and robustness to shortcuts.

### **Weakness**

The paper requires more benchmarks, in particular with causal learning methods. One proper benchmark would be the WILDS dataset [1]. Comparison with other methods such as [2] would be helpful.



[1] Koh et al.,  WILDS: A Benchmark of in-the-Wild Distribution Shifts
[2] Lee et al., Diversify and Disambiguate: Learning From Underspecified Data

---

> ### Author Response · Authors · 2022-08-02
> **Response to Reviewer aZBR**
>
> We are grateful for the supportive comments and kind words in your review and thank you for your remarks about the strengths of our method. As noted in your review, Chroma-VAE is well-motivated and novel. Besides our findings about the information bottleneck, we present a concrete method to use a generative model to tackle shortcut learning. Our quantitative experimental findings and qualitative visualizations across the board demonstrate the effectiveness of Chroma-VAE at mitigating shortcut learning.
>
> We refer you to our general response comment thread for a full response to the sole limitation mentioned in this review, which has to do with benchmarking. In particular, we explain why many of the datasets in WILDS are not applicable for the problem and context that we are trying to solve. We also explain why the comparison to Diversify and Disambiguate would not be a fair one, though we note that we have cited it appropriately in the Related Work section.
>
>
> **------------- Responses to specific questions -------------**
>
> **Q:** It is not clear why gender would be a shortcut and not the hair color, if both are correlated with the output labels.
>
> **A:** In this particular example, the preference for learning gender cues is likely an inductive bias of the network. Gender vs. blond hair is a benchmark task within the spurious correlation community and is commonly evaluated on [e.g. 2, 3].
>
> **Q:** In Figure 4. C, the dimensions of z_1 is more than z_2. Shouldn’t it be smaller since z_1 is correlated to shortcuts and thus more compressible?
>
> **A:** In general, the sizes of z_1 and z_2 depend on the complexity of the shortcut, as well as complexity of the overall generative modeling process. Here, we found the optimal value of z_p by hyperparameter searching.
>
> **Q:** What does ‘invariant’ means in Table2.
>
> **A:** Invariant refers to an equivalent model trained on grayscale MNIST, which is used as a reference for ideal performance, as done in [1].
>
> **Q:** “MLPs with smaller hidden layers have better accuracies, and our top performance is a k-nearest neighbor classifier with k = N^(½).” What does N denote here?
>
> **A:** N is the dataset size.
>
> [1] Martin Arjovsky, Léon Bottou, Ishaan Gulrajani, and David Lopez-Paz. Invariant risk minimization. 2019.
>
> [2] Shiori Sagawa, Pang Wei Koh, Tatsunori B Hashimoto, and Percy Liang. Distributionally robust neural networks for group shifts: On the importance of regularization for worst-case  generalization. 2019.
>
> [3] Evan Z Liu, Behzad Haghgoo, Annie S Chen, Aditi Raghunathan, Pang Wei Koh, Shiori
> Sagawa, Percy Liang, and Chelsea Finn. Just train twice: Improving group robustness without training group information. 2021.

---

> > ### Comment · Reviewer_aZBR · 2022-08-05
> > **Response to author rebuttal**
> >
> > > Q: It is not clear why gender would be a shortcut and not the hair color, if both are correlated with the output labels.
> > >A: In this particular example, the preference for learning gender cues is likely an inductive bias of the network. Gender vs. blond hair is a benchmark task within the spurious correlation community and is commonly evaluated on [e.g. 2, 3].
> >
> >
> > I agree that gender vs. blond hair is a standard test in the spurious correlation community; my understanding of Chroma-VAE is that shortcuts correspond to less complex/simple features and can be represented in a small fraction of the latent embedding of VAE. Gender is a more complicated feature than a hair color [1].
> >
> > >Q: In Figure 4. C, the dimensions of z_1 is more than z_2. Shouldn’t it be smaller since z_1 is correlated to shortcuts and thus more compressible?
> > >A: In general, the sizes of z_1 and z_2 depend on the complexity of the shortcut, as well as complexity of the overall generative modeling process. Here, we found the optimal value of z_p by hyperparameter searching
> >
> > Ideally, z_1 should be of low dimensions as shortcuts contain less information. For hyper-parameter fine-tuning, we would require a worst-group validation dataset limiting the impact of the proposed approach.
> >
> >
> >
> >
> > [1]. Scimeca et al., Which Shortcut cues will  DNNS choose? A study from the parameter space perspective.

---

> > > ### Author Response · Authors · 2022-08-09
> > > **Response to Reviewer aZBR (2)**
> > >
> > > Thank you for your reply and for taking the time to engage with our response. We would like respond to these two points below:
> > >
> > > > I agree that gender vs. blond hair is a standard test in the spurious correlation community; my understanding of Chroma-VAE is that shortcuts correspond to less complex/simple features and can be represented in a small fraction of the latent embedding of VAE. Gender is a more complicated feature than a hair color [1].
> > >
> > > Indeed, while it may seem intuitive that gender is more “complicated” than hair color and ought to be harder to learn, this does not appear to be the case for CelebA. Gender cues are in fact more easily picked up by the model — this is the reason why gender is such an “effective” shortcut when spuriously correlated to hair color and why CelebA is a useful benchmark experiment. Chroma-VAE operates on the _same_ principle here: because gender is preferentially encoded, it will end up being learnt by the z1 representation.
> > >
> > > As further empirical demonstration of this, we can consider the following experiment. We train and evaluate a simple ERM classifier with a heavily bottlenecked latent representation (z = 2) on gender and hair color labels separately:
> > >
> > > |                                            | Average acc. | Worst-group acc. |   |   |   |
> > > |--------------------------------------------|----------|------------------|---|---|---|
> > > | Trained and evaluated on gender labels     | 0.967    | 0.844            |   |   |   |
> > > | Trained and evaluated on hair color labels | 0.919    | 0.589            |   |   |   |
> > >
> > > As we can see, with limited model capacity, the model can predict gender better than hair color. In particular, the WGA takes a significant hit on hair color labels, precisely because the model prefers to encode gender features and not hair color features. This is the shortcut learning phenomenon at work.
> > >
> > > One possible hypothesis for this seemingly unintuitive result could be that as CelebA is a rather homogenous dataset (e.g. conventionally attractive celebrities with more limited racial diversity), the model ends up simply learning a narrow and simple facet of gender (e.g. presence of lipstick) that is, by itself, less complex and more easily picked up than hair color. Such a trait might not be present in a more diverse dataset like the UTKFace dataset considered in the paper you linked (Scimeca et al., 2022).
> > >
> > >  > Ideally, z_1 should be of low dimensions as shortcuts contain less information. For hyper-parameter fine-tuning, we would require a worst-group validation dataset limiting the impact of the proposed approach.
> > >
> > > We have empirically observed that tuning via _average_ accuracy (without using group labels) suffices: that is, the hyperparameter setting that gives the best average accuracy on the validation data also performs the best on WGA. We hypothesize the reason for this observation in Lines 194-202 of the paper: We note that the effectiveness of Chroma-VAE depends on (i) the shortcut being preferentially encoded and successfully isolated in z1 and (ii) z2 having sufficient capacity to learn the non-shortcut predictive features. In practice, (ii) is achieved by choosing dim(z) appropriately to ensure good reconstruction — this can be evaluated without group labels, e.g. validation ELBO or visual inspection. Given such an appropriate dim(z), most non-extreme values of dim(z1) and dim(z2) will ensure that z1 has enough capacity to encode the shortcut — indeed, our experiments were carried out with a very sparse search on the hyperparameter space.
> > >
> > > **--------------------------**
> > >
> > > We hope that our question-specific responses here and our general response above has answered some of the concerns or areas of ambiguity you might have about the paper. We reiterate that our results across the various benchmark experiments (CelebA, Colored MNIST, Dominoes, CXR) as well as supporting visualizations (e.g. partial reconstructions) has shown that Chroma-VAE can learn shortcut-invariant representations, helping to mitigate the harmful effects of shortcut learning. We hope that you can consider raising your score if you are satisfied with our response.

---

> > > > ### Comment · Reviewer_aZBR · 2022-08-09
> > > > **Response to author rebuttal (2)**
> > > >
> > > > 1. Thank you for presenting new results on gender vs. hair color cues. I find this interesting and hope that the discussion will be useful to the readers for future works. I have updated my score.

---

### Official Review · Reviewer_qobE · 2022-07-12

**Rating:** 7
**Confidence:** 4
**Soundness:** 1 poor
**Presentation:** 2 fair
**Contribution:** 2 fair

**Summary:**

The authors present an approach that aims to prevent "generative classifiers" from learning spurious correlation, which the authors refer to as short-cuts. They achieve this by splitting the latent representation in two, z1 and z2. The latent, z1, is passed to a classifier, while z1 and z2 are used to reconstruct the image. The model is trained using gradients from the classifier and the ELBO loss. The motivation being that z1 should contain the short-cut info, while z2 should not.

**Questions:**

The authors describe short-cut learning as “using simple features to achieve low training loss”. Surely it is best for the model to learn the simplest explanation since this is more likely to generalise? It would be helpful here to make the connection between “short-cuts” and spurious correlations.

Line 39-40: By “generative classifier” do you mean a classifier trained on representations learned via a generative model? Could you provide some (anecdotal) example of this?

To what extend is short-cut learning a failure to collect sufficiently diverse and unbiased data?

Line 90-95: In an unsupervised representation learning setting there are no labels, do these short-cuts still appear here? And if not, why not just fine-tune a classifier on top of representations learned without supervision?

Figure 2(a): Reconstructions look poor in all cases. The do **not** appear to be especially bad in the case with the blue square.

Figure 2(b): These results are very interesting and do motivate the problem well. This figure could also be improved by clearly showing an x-axis with the model capacity increasing and a box around the square.

This approach can only be used to factor out the information specific to the classes you have labels for?

In Figure 3: what is the class label here? Is it hair colour? Please clarify in the caption.

In Figure 3(d)-(e) please show the original faces too. This figure does show that this approach is “working”. However, it also suggests that gender is being encoded? Is there bias in the data to blonde women?

Table 2: Is it worth pointing out that the authors expect z1-clf to be poor on D_out? And that this further goes to show that z1 contains the spurious correlations?

It is not clear to me how using z1 for classification helps to ensure that z2 is better than a representation learned using a VAE without supervision (other than the fact that z1 and z2 share parameters, but to me this does not appear to be sufficient)? Is this achieved because z1 and z2 are needed for reconstruction and the assumption is that the information would not need to be encoded in both z1 and z2? This is very unlikely to hold in most cases. It may hold for a beta-VAE (or other disentangled representation). However, then there may not be sufficient information in z2 for classification?

Does the approach only work for single label classification?

Table 3 and 4: why is there such high variation in accuracy between groups?

Table 4: Include the dataset information in the caption.

**I am happy to significantly increase my score if the authors can address the following:**
(1) How does the current setup explicitly prevent "short-cut" information being encoded in z2.
(2) Add a clear diagram with losses and gradients.
(3) It is highly commendable that the authors include results on real world data and I do not want to discourage this. Can the authors please explain any benefits their approach is having on this dataset or why we should be optimistic about the results.

**Limitations:**

Limitations are not addressed in the main body of the paper.

**Strengths And Weaknesses:**

Originality:
This work proposes using a classifier on only a sub-set of the latent representation rather than the whole representation.

Quality:
It is very encouraging and highly commendable that the authors show results on a real world dataset, this is excellent! While their approach does much better on worst group accuracy, it still does very poorly on the average accuracy. Can the authors please elaborate more on this and explain why this result is still encouraging? Or if this suggests that their approach is not suitable for real world problems? Why do the other methods perform better?

It is not clear that there are sufficient pressures to ensure that the short-cut is not also encoded in z2, especially in non-trivial settings. This may explain why the model performs poorly on real world data.

There is very high variance between accuracy on different groups of data.

Clarity:
Aside from some terminology confusions mentioned in the box below, this work is exceptionally well motivated in the introduction.

It would be very helpful to have a diagram showing the encoding, decoding, classification with losses and gradients. The exact implementation and gradients are not 100% clear.

Significance:
The authors tackle an important and challenging problem. However, they fail to show convincing results on real world data.

Main strengths:
(1) The work is very well motivated.
(2) The authors evaluate on real-world data.

Main weaknesses:
(1) It is not clear how the approach works to mitigate short-cut information being encoded in z2.
(2) The approach does not work well on real-world data.
(3) The positive results are on simple datasets.

---

> ### Author Response · Authors · 2022-08-02
> **Response to Reviewer qobE**
>
> We are grateful for the time that you took in reviewing our work. As noted in your review, Chroma-VAE is novel and well-motivated. By partitioning the latent space, Chroma-VAE successfully learns shortcut-invariant representations in one of the two latent subspaces, which we empirically demonstrate using reconstruction visualizations. We believe that our work represents an unique way to tackle the shortcut learning problem and has value to both the shortcut learning and generative modeling communities.
>
> Here, we would like to address the three main questions posed in your review regarding the real-world effectiveness of Chroma-VAE.
>
> **(1) How does the current setup explicitly prevent "short-cut" information being encoded in z2.**
>
> Firstly, in Chroma-VAE’s setup, both z1 and z2 are used for image reconstruction. The VAE is therefore disincentivized to encode the same features in z1 and z2, especially under limited latent capacity (which we can control as a hyperparameter), because that will harm the quality of reconstruction and lead to higher training loss. Since z1 preferentially encodes the shortcut features due to the initial classifier head being attached **only** to z1, the model therefore exerts pressure on z2 to encode other salient features for reconstruction.
>
> For this mechanism to work effectively, we only need to ensure that z1 and z2 are capable of encoding different features of the image without duplication. To this end, we note that in our experiments, we observed that one simple trick is having separate feature maps for the z1 and z2 encoders (without parameter sharing), which effectively permits z1 and z2 to learn different features.
>
> Finally, whether shortcuts have been leaked to z2 is something that we can ultimately confirm visually, using partial reconstructions.  Indeed, in all the experiments in our paper, we have empirically demonstrated that the partial reconstructions of z2 are invariant to the shortcut feature, i.e. different samples of the partial reconstructions will contain different “traits” of the shortcut feature. E.g. in the FashionMNIST-MNIST example, the z2 reconstructions contain both 0 and 1 digits.  For a new dataset that Chroma-VAE is deployed on, we can similarly confirm the efficacy of the method using partial reconstructions.
>
> **(2) Add a clear diagram with losses and gradients.**
>
> We have added this diagram — see Figure 9 in Appendix D of the updated paper draft, which shows the entire training procedure for Chroma-VAE, with associated loss terms and backpropagated gradients.
>
> **(3) It is highly commendable that the authors include results on real world data and I do not want to discourage this. Can the authors please explain any benefits their approach is having on this dataset or why we should be optimistic about the results.**
>
> In general, X-ray data is difficult to model and reconstruct. Existing classifiers on CXR datasets use ImageNet pre-trained models entirely [e.g. 1],  and to the best of our knowledge, there is no work out there that use VAEs to _generatively_ model CXR data. Since our method is VAE-based and requires the model to reconstruct well, we believe that the inherent difficulty of training the VAE architecture is responsible for the poorer average accuracy, not a failing in the principle of our method. For example, we had to use a pre-trained ImageNet encoder paired with a deconvolutional decoder. (We found that a CNN-based encoder fared even worse.)
>
> Despite the difficulty and challenge of modeling X-ray data, we ultimately chose to include this experiment for two reasons:
>
> (1) Firstly, there are very few meaningful real-world datasets for shortcut learning that we can use. Indeed, most existing work in this field choose to evaluate entirely on benchmarks like CelebA [e.g. 2, 3, 4]. CXR was the best example of a real-world dataset where shortcut learning is present.
>
> (2) Furthermore, our primary goal for this experiment is not to be competitive with state-of-the-art, but to show that the principle of our approach works on real-life data, not merely “synthetic” and easily-modeled datasets like MNIST. Indeed, despite the poorer average accuracy, our method still beat the other baselines in terms of WGA. Indeed, this shows that the methodology behind Chroma-VAE is fundamentally sound and does successfully mitigate the network from learning shortcuts.
>
> We hope that our responses have adequately addressed your concerns about real-life applicability. In particular, we hope that our responses to (1) and (2) show that Chroma-VAE’s methodology is fundamentally sound — a claim also backed up by our experimental findings, both quantitative (WGA) and qualitative (reconstruction figures). We would appreciate if you consider raising your score in light of these responses.

---

> > ### Comment · Reviewer_qobE · 2022-08-08
> > **Thanks for your response.**
> >
> > (1) It is now more clear to me how the model is intended to work. A more clear demonstration that z2 does not contain any short cut features would be to show that you cannot train a classifier on those representations (i.e. that you cannot get a classifier to overfit)  to predict the information). Visual inspection is not sufficient on its own.
> >
> > (2) Thank you for adding the figure.
> >
> > (3) Again, it's great that the authors demonstrate their results on real world data. For this reason I am happy to increase my score to a 6 with the expectation that the authors will conduct the experiment suggested in (1).

---

> > > ### Author Response · Authors · 2022-08-09
> > > **Response to Reviewer qobE (2)**
> > >
> > > First of all, we thank you for taking the time and effort to read and digest our detailed response. Indeed, we can explicitly demonstrate the final follow-up point you raised in (1). We propose an experiment as such: after having learned the initial VAE and z1-classifier head, we subsequently train and evaluate the z2 representation on **shortcut labels** instead of the true labels.
> > >
> > > If Chroma-VAE can truly prevent shortcut information from leaking to z2, then the z2-classifier should not be expected to do well, not even on _average_ accuracy. We performed this experiment using the same two datasets (CelebA and FashionMNIST-MNIST) as Section 5.2 of our paper, on the hard 100% correlation setting and measuring test average accuracy. For reference, we also show the results where z2 is trained and evaluated on the true labels (the intended purpose), using the same hyperparameter settings for both sets of results.
> > >
> > > |                       | z2-classifier on shortcut labels | z2-classifier on true labels |
> > > |-----------------------|--------------------------------------------------------|----------------------------------------------------|
> > > | CelebA (Gender/Blond) | 0.580                                                  | 0.820                                              |
> > > | FashionMNIST-MNIST    | 0.338                                                  | 0.881                                              |
> > >
> > >
> > > As can be seen from the table above, the z2-classifier performs much more poorly when trained and evaluated on shortcut labels directly. These results therefore show that the z2 representation safely does not contain the shortcut feature.
> > >
> > > **--------------------------**
> > >
> > > With this additional experiment, we hope that we have addressed your concerns about the soundness and validity of our approach to avoid shortcut learning. As we have shown, our method is effective on various popular benchmark datasets. Furthermore, despite the difficulty of modeling real-world CXR data, which most papers in this space do not even attempt, we show that our method is indeed sound and raises WGA significantly. In light of this, we hope that you would consider raising your score.

---

> ### Author Response · Authors · 2022-08-02
> **Response to Reviewer qobE (continued)**
>
> **------------- Responses to specific questions -------------**
>
> **Q:** The authors describe short-cut learning as “using simple features to achieve low training loss”. Surely it is best for the model to learn the simplest explanation since this is more likely to generalise? It would be helpful here to make the connection between “short-cuts” and spurious correlations.
>
> **A:** It is not necessarily true that the simplest feature generalizes well. One counter-example is traffic classification in self-driving cars, where the background (day or night) is a shortcut that is much simpler than the true classes that we want to learn (pedestrian, vehicle, traffic light, etc.), yet nightime can indicate that pedestrians are unlikely to be on the road.
>
> By “shortcut”, we mean features that are not only spuriously correlated, but also simple and highly compressible. As mentioned in our general response comment thread, it is the combination of being strongly correlated and easily encoded that induces networks to prefer learning shortcuts in lieu of salient features. Shortcut learning is a problem precisely because the simplest feature is often *not* what we want to learn.
>
> **Q:** Line 39-40: By “generative classifier” do you mean a classifier trained on representations learned via a generative model? Could you provide some (anecdotal) example of this?
>
> **A:** Yes. As mentioned in the Related Work section, class-conditional models and hybrid models represent the two main ways of using representations learned via a generative model to classify. For example, in a class-conditional model, we train one generative model per class present in the training data. When given a new data point, we compute each class likelihood p(x|c) and use Bayes’ Rule to make the final classification.
>
> **Q:** To what extend is short-cut learning a failure to collect sufficiently diverse and unbiased data?
>
> **A:** While having a more diverse and unbiased dataset can mitigate shortcut learning, it is not always realistically possible or practical to do. It can be difficult to collect data in certain situations, e.g. each hospital has limited patient data and has patients sampled from a distribution different from other hospitals.
>
> **Q:** Line 90-95: In an unsupervised representation learning setting there are no labels, do these short-cuts still appear here? And if not, why not just fine-tune a classifier on top of representations learned without supervision?
>
> **A:** Without labeled supervision, there is no guarantee that salient non-shortcut features will be captured in the learned representations.
>
> **Q:** Figure 2(a): Reconstructions look poor in all cases. The do not appear to be especially bad in the case with the blue square.
>
> **A:** Note also that in the examples where the blue patch shortcut happens, the reconstructions all look quite similar to each other despite the original images being different (mode collapse), further confirming that the latent embedding is mostly encoding the shortcut.
>
> **Q:** Table 2: Is it worth pointing out that the authors expect z1-clf to be poor on D_out? And that this further goes to show that z1 contains the spurious correlations?
>
> **A:** Yes and yes.
>
> [1] Aahlad Puli, Lily H. Zhang, Eric K. Oermann, Rajesh Ranganath. Out-of-distribution Generalization in the Presence of Nuisance-Induced Spurious Correlations. 2022.
>
> [2] Shiori Sagawa, Pang Wei Koh, Tatsunori B Hashimoto, and Percy Liang. Distributionally robust neural networks for group shifts: On the importance of regularization for worst-case  generalization. 2019.
>
> [3] Evan Z Liu, Behzad Haghgoo, Annie S Chen, Aditi Raghunathan, Pang Wei Koh, Shiori
> Sagawa, Percy Liang, and Chelsea Finn. Just train twice: Improving group robustness without training group information. 2021.
>
> [4] Yoonho Lee, Huaxiu Yao, and Chelsea Finn. Diversify and disambiguate: Learning from underspecified data. 2022.

---

### Official Review · Reviewer_x5WA · 2022-07-13

**Rating:** 7
**Confidence:** 3
**Soundness:** 3 good
**Presentation:** 3 good
**Contribution:** 3 good

**Summary:**

This paper proposes a method to mitigate the pervasive shortcut learning issue. The authors first make the observation that deep generative classifiers tend to encode spuriousness when the bottleneck is present. Based on this insight, authors propose a two-pronged approach where a generative classifier is trained while only a partition of the latent space is used for classification, and this part of the latent space absorbs the spuriousness. In the second stage, only the other latent code/representation is used to predict the label.

**Questions:**

- In the paper, the observation that the presence of the bottleneck affect how much the spuriousness will be picked up. Can this be shown as an ablation for the main experiments that when the z1-clf has larger bottleneck capacity (i.e., varying $z_p$), it carries less spuriousness and thus results in worse OOD performance?

**Limitations:**

Yes.

**Strengths And Weaknesses:**

Strengths
- The paper tackles an important long standing problem. The motivation is intuitive and the method is simple and easy for the community to reproduce and build on top.
- The paper is laid out clearly and easy to follow, the logic flows smoothly
- Empirical results show large benefit over the chosen baselines
- The analyses are illuminant

Weaknesses
- The choice of baselines seems a bit slim. The main baselines are the vanilla ERM and the generative classifier. The only stronger baseline was JTT. The results will be more convincing if put into context w.r.t. other recent work that tries to mitigate spuriousness such as NURD (Out-of-distribution Generalization in the Presence of Nuisance-Induced Spurious Correlations; https://openreview.net/pdf?id=12RoR2o32T).

---

> ### Author Response · Authors · 2022-08-02
> **Response to Reviewer x5WA**
>
> We are very grateful for your supportive review and kind words towards our work. As noted in your review, Chroma-VAE makes novel use of a generative model to tackle shortcut learning, which is an important and pertinent challenge that deep networks encounter. Both our quantitative experimental findings and qualitative visualizations demonstrate the effectiveness of Chroma-VAE at mitigating shortcut learning.
>
> We refer you to our general response comment thread for a full response to the sole limitation mentioned in this review, which has to do with benchmarking. In particular, while we have considered NURD and indeed cited it as related work, we explain that a comparison to NURD would be unfair as NURD makes use of group labels, which Chroma-VAE does not require.

---

### Author Response · Authors · 2022-08-02
**General Response**

We thank our reviewers for their insightful and supportive remarks, and we appreciate their positive reception towards the novelty and relevance of our work. As noted by all reviewers, our work is novel, being the first to explore shortcut learning from the lens of generative classifiers. Besides establishing a connection between shortcut learning and the information bottleneck, we present a new approach for learning a shortcut-invariant input representation useful for supervised downstream tasks.

We believe that our work is a timely and valuable contribution to the shortcut learning community, one that is sufficiently different from and complementary to existing approaches. Furthermore, our work is also of broad value to the generative modeling community, by revealing a significant application area, beyond image sampling, where generative classifiers show promise.

In addition to the specific responses to each reviewer, we wish to address two main themes here.

**(1) Chroma-VAE focuses on features (shortcuts) that are preferentially encoded by networks.**

A key finding of our work is that shortcuts are preferentially encoded by the network. This is due to a combination of two factors: (i) shortcuts are strongly correlated with the label (therefore learning them results in low training classification error), and (ii) shortcuts are easily compressible and simple to learn (therefore favored by the model’s inductive biases).

As such, shortcuts are not just spurious correlations — they are spurious features that models prefer to learn, especially under limited model capacity. Our work echoes the findings of other work like [1], which note the “simplicity bias” of such neural networks.


**(2) Benchmarking and Baselines [x5WA, aZBR]**

We have thought carefully about the benchmark datasets and methods that can be fairly compared to our work. We note that Chroma-VAE is designed to specifically address shortcut learning, instead of distribution shifts in general. As such, we chose benchmark datasets where such shortcuts do exist. Many of the WILDS [2] datasets are not applicable as they do not have a shortcut feature present. Those that do (e.g. CivilComments, as used in JTT) are text datasets which Chroma-VAE cannot model.

A key motivation of our work is to present a method that does not require additional sources of data that might be unrealistic in many real-life situations. We did not implement methods where such additional sources of data are necessary, since the comparison would not be fair. For example, NURD [3] is not a fair comparison as it requires group labels for the training data, which we do not use. Similarly, DivDis [4] requires unlabelled test data (presumed to be balanced across groups) to learn a diverse set of functions.



[1] Harshay Shah, Kaustav Tamuly, Aditi Raghunathan, Prateek Jain, Praneeth Netrapalli. The Pitfalls of Simplicity Bias in Neural Networks. 2020.

[2] Pang Wei Koh, Shiori Sagawa, Henrik Marklund, Sang Michael Xie, Marvin Zhang, Akshay Balsubramani, Weihua Hu, Michihiro Yasunaga, Richard Lanas Phillips, Irena Gao, Tony Lee, Etienne David, Ian Stavness, Wei Guo, Berton A. Earnshaw, Imran S. Haque, Sara Beery, Jure Leskovec, Anshul Kundaje, Emma Pierson, Sergey Levine, Chelsea Finn, Percy Liang. WILDS: A Benchmark of in-the-Wild Distribution Shifts. 2020.

[3] Aahlad Puli, Lily H. Zhang, Eric K. Oermann, Rajesh Ranganath. Out-of-distribution Generalization in the Presence of Nuisance-Induced Spurious Correlations. 2022.

[4] Yoonho Lee, Huaxiu Yao, and Chelsea Finn. Diversify and disambiguate: Learning from underspecified data. 2022.

---

### Meta-Review · Area_Chair_Hqes · 2022-08-31

**Recommendation:** Accept
**Confidence:** Certain

**Metareview:**

This paper explores a method to learn generative classifier while mitigating 'shortcut learning'---relying on spurious correlations. They do so by modelling two latent variables, only one of which is used to classify, and both used to generate data. The underlying intuition is that 'shortcuts' will only manifest in a small fraction of the latent space embedding as they compress. Experiments show the efficacy of the method on multiple datasets.

The reviewers all agreed that the paper tackles an interesting and relevant problem, that the exploration of the performance and experiments considered are quite thorough.

The biggest issue raised in this instance is that the experiments largely only deal with restricted/constrained settings. While it is understandable that the problems tackled here are well accepted benchmarks within this subfield, the paper should make clear it's contribution and include a discussion about the potential for real-world application.
In particular, the X-ray example was something that was highlighted as something that, as currently presented, potentially can imply that the proposed method is viable for actual real-world use. It is strongly suggested that the authors moderate the interpretation / claims / discussion around this example.

The authors also provided additional experiments over specific questions on the information in the non-classifier latent to address reviewer concerns, which was good.

Overall the paper has merit and explores an interesting idea with a good set of experiments that cover a range of settings, and should be accepted.


**Award:**

No

---

### Decision · Program_Chairs · 2022-09-14

Accept